# Branding4Resilience: Explorative and Collaborative Approaches for Inner Territories

**Maddalena Ferretti** [1,*], **Sara Favargiotti** [2], **Barbara Lino** [3] **and Diana Rolando** [4]

1 Department of Construction, Civil Engineering, and Architecture, Università Politecnica delle Marche, 60131 Ancona, Italy

2 Department of Civil, Environmental and Mechanical Engineering, Università degli Studi di Trento, 38123 Trento, Italy

3 Department of Architecture, Università Degli Studi di Palermo, 90128 Palermo, Italy

4 Department of Architecture and Design, Politecnico di Torino, 10125 Turin, Italy

* Correspondence: m.ferretti@univpm.it

**Abstract:** This article analyzes inner and marginal territories in four Italian peripheral contexts by first discussing some of the results and future steps of the "B4R Branding4Resilience" research project, funded by the Italian Ministry of Research from 2020 to 2023. The overall research is based on three phases: (1) the exploration phase to analyze socio-economic data and territorial dynamics; (2) the co-design phase involving local actors to develop ideas for a selected pilot case; (3) the co-visioning phase where a future transformative perspective for the whole area was shared with the institutions. The article focuses on phase 1 and presents some first results achieved by the application of a methodological approach based on the integration of different qualitative and quantitative tools and methods. The results outline the exploration of the four selected territories through data analyses and mapping, perceptive-narrative explorations, field research, and explorative designs. The concept of peripherality is addressed in a critical way, trying to go beyond standardized definitions, including interdisciplinarity as an essential tool for territorial enhancement and branding. The main interpretation findings not only outline possible strategies and actions for the four analyzed inner territories, but also foster the application of the proposed methodological approach in other complex socio-economic contexts.

**Keywords:** branding; resilience; inner territories; habitat; territorial enhancement; design; architecture; landscape; urbanism; real estate appraisal

## 1. Introduction

The issue of the growing marginalization of territories with different characteristics and localizations is common in many European regions, where the gap between the development dynamics of catalyzing urban formations and the progressive depopulation and abandonment of marginalized areas is evident and is getting wider and more polarized [1–6].

The progressive impoverishment and depopulation of large portions of the territory and the worsening of territorial inequalities are also, in Italy, the result of the concentration of settlement processes around the main central urban areas, in those inner territories in which the greatest efforts of public policies aimed at technological, infrastructural and service innovation have been fostered for years [7]. Despite the efforts made by the Italian National Strategy for Inner Areas (SNAI) both to promote the debate in the field and to propose operational tools to counteract the processes of territorial marginalization [8], the importance and the urgency of studying inner territories' potentials have been recently relaunched by the dynamics that the COVID-19 pandemic and ensuing emergency brought out.

On the one hand, the experience of the COVID-19 pandemic rekindled the debate on inner areas nourishing rhetorical positions on the beauty of spread settlements and villages and on life in small towns as opposed to urban life. This approach has flattened awareness of the transformational needs that inner areas have to face, even more today than before [9]. On the other hand, the COVID-19 pandemic strengthened the perception of those territorial inequalities between metropolitan and inner areas that have been worsened by the crisis [10]. Inner areas are also poorly connected to technological infrastructure networks, where, instead, a very good digital connectivity is necessary to compensate for the distance from services (see, for example, the Italian Ultra-Broadband Strategic Plan—https://bandaultralarga.italia.it/en/strategia-bul/strategia/, accessed on 1 August 2022).

In this context, the attention to the study of the territorial and socio-economic dynamics that characterized the inner areas and their evolution during and after the COVID-19 pandemic is evident. However, it is worth remembering that public administrations in these fragile territories need to be supported by local governments and academics, in order to deeply know the past and ongoing trends that could undermine the territorial development, as well as the potential strengths that could be enhanced. In the scientific debate, different tools and methods support processes aimed at the exploration of territories and the proposal of design interventions [11–14], but for their implementation a clear and replicable methodological approach is essential to correctly face and share strategies and actions.

The researchers have amply demonstrated that the impacts of the emergency have strongly highlighted the numerous contradictions of being at the margins, opening up spaces of opportunity also for inner areas [15], and the relevance of territorial policies in which the margins find a renewed centrality [16,17]. However, it is also evident that the road opened by SNAI must be pursued and even overcome both in terms of reflection on the classification criteria. There is also the need to deepen the peculiarities of the different trajectories and problems of these territories, but also in compliance with the operational reactivation processes, especially with a view to the increasingly active involvement of relevant actors in innovation paths [9].

Therefore, this article proposes a methodological approach aimed at integrating different qualitative and quantitative tools and methods to explore inner territories in-depth, interact with local authorities and stakeholders of the territories, and support them in the definition and sharing of enhancement projects. Moreover, this article aims to reflect on the future of inner territories in the Italian context by testing the proposed methodological approach and outlining and discussing the first explorative results achieved in the context of "B4R Branding4Resilience. Tourist infrastructure as a tool to enhance small villages by drawing resilient communities and new open habitats" (B4R), a research project of national interest (PRIN 2017—Young Line). B4R was funded by the Italian Ministry of University and Research (MUR) for a three year duration (2020–2023). The project is coordinated by the Università Politecnica delle Marche (principal investigator Maddalena Ferretti) and it involves the Università degli Studi di Palermo (local coordinator Barbara Lino), the Università degli Studi di Trento (local coordinator Sara Favargiotti), and the Politecnico di Torino (local coordinator Diana Rolando) as partners. In this article, the results of the first exploratory phase of the B4R project are presented.

The article is structured in four main parts. A background and project framework provides the reference for the context of Italian inner areas, attempting to go beyond the current interpretations and positions. The next section explains the methodological approach that has been applied in the context of the national research project, focusing on its main initial step. The third part of the article summarizes the results achieved by means of the integrated application of some methods and tools (data analyses and mapping, stakeholder analysis, a collaborative platform, as well as some tentative territorial portraits to devise the main characteristics and findings of the areas). Finally, the discussion focuses on the different topics emerging from the results, sketching project themes for the branding operative actions that are to be implemented in the territories in the next research phases.

## 2. Background and Project Framework

About 58% of the European population live and work in "rural areas" and in "towns and suburbs" that are respectively considered "thinly populated areas" and "intermediate density areas" [18]. Disparities are still evident at the regional level across the EU [2] and the complexity of territorial imbalances and spatial inequalities has been highlighted in the literature focusing on deprived, declining, and marginalized areas. A multitude of terms referred to gaps between core areas and areas suffering from territorial imbalances are used in many multidisciplinary debates: peripheral area, marginal area, inland area or inner area [1]. Peripheralization processes are described from the perspective of social inequalities theories as linked to the production of socio-spatial disadvantages. Other authors in the field of political theories connect peripheralization to conditions of exclusion [19]. Many other researchers look at this issue focusing on regional innovation strategies under EU cohesion policy and evaluate its limits with a critical perspective [3,4]. The European operative programs GEOSPECS [20] and PROFECY [21] define different types of inner peripheries according to social, economic, and spatial imbalances and only according to quantitative indicators connected to distance from regional centers, lowered economic capacity, number of accessible services, and presence of impoverished areas. In Italy, the issues of territorial imbalances and spatial inequalities were translated into an overarching national-level planning instrument for counteracting the marginalization and demographic decline of Italian inner areas: the National Strategy for Inner Areas (SNAI—https://www.agenziacoesione.gov.it/strategia-nazionale-aree-interne/?lang=en, accessed on 1 August 2022) [8,22,23]. According to this strategy, those areas that are marginal and disconnected are defined as "inner". All over Italy—from the Alps, across the Apennines, and to the islands—they are not residual, but account for almost 53% of municipalities, hosting 23% of the population—nearly 13,540 mil. people—and covering about 60% of the entire territory [24].

While SNAI's Italian experience represents an important test-field for operational experimentation and methodological innovations, there are also some obvious limitations and a number of implementation problems have surfaced. For example, the need to also focus on peripheral areas that are outside the areas covered by SNAI or to go beyond parameters that lead to standardizing and normalizing complex phenomena [9] and to overcome the opposition between top-down and bottom-up actions, reinterpreting the place-based approach [25]. However, the SNAI strategy has recently benefited from new funds from the 2021 National Recovery and Resilience Plan [26] and the 2021–2027 European Cohesion Policy, which aim to counteract the marginalization and demographic decline of the Italian inner areas. Assuming the SNAI framework, several kinds of research were developed in the Italian context in order to analyze the opportunities and limitations of those rural depopulated areas distant from the main service centers of education, health, and mobility. Battaglia et al. [12], for example, assumed inner areas as a specific territorial weak category and proposed a territorial planning model of local development based on the implementation of multi-disciplinary approaches and the involvement of local stakeholders in planning choices to reduce marginality and stimulate the local economy. The classification proposed by SNAI was also adopted by Urso et al. [13], who analyzed the potential structural changes of non-inner vs. inner areas and assessed their adaptive capacity to the consequences of the 2007–2008 economic crisis. Their study highlighted that inner areas and urban poles differently re-adapt their local economies by demonstrating their future resilience. The potential contribution of local enterprises to the growth of Italian inner areas by means of the creation of partnerships was investigated by Mastronardi and Romagnoli [11], who highlighted that these new firms may lead to real and enduring benefits to local communities, as well as contribute to the inversion of the demographic decline.

Within this framework, this research proposes to shift the concept from inner areas to inner territories embracing those contexts affected by depopulation, peripherality, aging, and lack of services and infrastructure [23], as fragile and hybrid spaces with high territorial complexity, both at the level of spatial development strategies and of governance issues.

Inner territories are gaining deeper consideration in debates, research, and practice all over Europe. They are hybrid contexts intended as territories that need to raise awareness of their hidden qualities [27], to address the conceptualization of peripheries as places of spatial opportunities and social innovation [28] and develop branding strategies based on their rural identity and perceptive components of authenticity [5,6]. Inner territories are spaces of opportunity that ask for a processual and multi-sectoral approach able to keep together different interests, conflicts, but also potentials, and resources. Putting into value territorial capital and local resources is a decisive success factor for these places, especially by recycling abandoned buildings or underused heritage. Very often in small villages, there are a great availability of obsolete and neglected buildings that represent a criticality in terms of maintenance costs for administrations and owners. The redevelopment of these abandoned or underused properties by means of the identification of new functions represent a great opportunity to create new tourist infrastructures, new spaces for public services, and foster new residential incentives [29].

In the context of European territorial cohesion policies, tourism is seen as a way to enhance territorial capital and is proposed as a valid driver to revitalize marginal territories [30]. Similarly, SNAI itself considers tourism as one of the five lines of local development, along with four other strategic assets. At the Italian national level, years after the SNAI started its investigation, the Ministry for Cultural Heritage and Cultural Activities and Tourism (MIBACT) has elaborated specific guidelines [31] that show how in inner areas tourism still plays a modest role compared to the stock of latent territorial capital. If mass tourism requires complex infrastructures, new forms of tourism more rooted in the communities, such as relational tourism [32], could play a fundamental role in the strategies of local development [31] with a global impact on the local economy, on the reactivation of under-utilized latent heritage, on social cohesion, and on communities' resilience. In particular, the theme of slow tourism in reference to marginal areas has been investigated both in Italy [33,34] and in the international literature [35], but there is no lack of awareness of the fact that for marginal territories, tourism cannot represent a pre-established recipe [36] and that it is rather necessary to activate virtuous and structural processes of regeneration that are rooted in local specificities [9,24].

Both in the Italian and international context, the experience of the COVID-19 pandemic rekindled the debate on inner areas and focused the attention on other possible development opportunities beyond tourism. Some of the recent debates tend to correlate often congested urban formations with an increased "susceptibility to infectious disease", as opposed to peripheral and marginal territories, where this dynamic seems reduced ([37], p. 246); these positions contributed to nourish anti-urban utopias. The goal of self-sufficiency and safety encouraged "escapes from the city" and fostered a radically opposed vision between inner and metropolitan areas. This approach is supported by a certain romanticism which at times led to rhetorical positions on the beauty of spread settlements and villages, and on life in small towns as opposed to urban life. At the same time, this approach lacks awareness of the transformational needs of inner areas [9,38,39]. Indeed, the COVID-19 pandemic strengthened the perception of those territorial inequalities and highlighted in all its severity the territorial differences, in terms of health and education services, in which the inner areas are by definition lacking. In addition, it highlighted the limits of the incessant urbanization and concentration of settlements, as well as the preponderance of infrastructure policies in large conurbations rather than in small municipalities [10].

*2.1. The B4R Project Framework*

Within the general background highlighted in the previous paragraphs, the framework of the B4R project (www.branding4resilience.it, accessed on 12 August 2022) helps to clarify the reference context for this article. B4R aims to define a development path for fragile inner territories through new impulses to tourist infrastructure for more resilient communities and open habitats [40]. The combination of branding and resilience outline the project's innovative contribution: using branding to foster tourism as an impulse for structural

transformation of territories, focusing especially on the reactivation of built heritage and unused building stocks. This would strengthen local economies and favor re-settlement processes in small, depopulated villages.

Against the backdrop of research and practices all over Europe, fragile territories are considered territories of opportunities. B4R contributes to this change of perspective, beyond the SNAI, connecting with the potential of tourism, but also advocating knowledge progress with regard to built heritage, landscape, planning, and enhancement of small villages and inner territories. Indeed, B4R explores a common European trend proposing a renewed role of peripheries as motors of innovation and test fields for new dynamics of development, connected to space, settlements, and landscapes. Through a branding strategy which is shared with local actors and communities, material and immaterial resources are connected to forms of relational tourism with the implementation of minimal infrastructures. Targeting the reactivation of places through small design interventions (e.g., community hubs) the project aims to foster social innovation initiatives and accelerate community resilience. Indeed, in B4R, the proposed idea of branding revolves around the powerful and disruptive capacity of design and spatial transformations to trigger development. Also, the project aims to attract new residents by addressing strategic scenarios [41] and accompanying administrations in the formulation of supporting policies [27,28]. Tourism, through branding, is therefore only the starting engine that can act as a complement and a multiplier of impacts to generate more rooted dynamics of change that mainly regard communities and places.

Through the exploration around and inside four Italian territories that face conditions of marginality but at the same time show positive impulses towards regenerative processes, the project proposes a new understanding and perspective towards more resilient futures. The branding strategies and operative actions reflect a double view, from the inside through the participation of communities, and from the outside through the increase of the areas' attractiveness. The final aim is to start new cycles and virtuous pathways of development to produce more structural changes and ultimately to increase the resilience of local communities that have contributed to address the transformations through shared and collaborative methods.

### 2.2. Overall Project Structure

B4R is based on three main phases, each corresponding to a research work package:

1. Exploration;
2. Co-design;
3. Co-visioning.

The "Exploration" phase, implemented in the first year of the research—and specific focus of this article—served to investigate the territorial contexts from different points of view, with a focus on spatial interactions. It is not based on a codified methodology, but it combines and integrates different quantitative and qualitative tools and methods, thus proposing a unique methodological approach. In this sense, it builds upon and tries to go beyond the methodological approach used by some of the authors in the transdisciplinary "Regiobranding" project [42]. The second phase is the co-design with communities, with the goal of proposing useful transformations of small infrastructures and operative branding actions in selected pilot cases. The third phase is the development of co-visioning processes in collaboration with local actors and institutions.

All the three phases highlight a multidisciplinary approach and the need to reposition design as a knowledge producer and a core discipline in urban and territorial development by means of collaborative methods [43–45]. The P.I. and the other three local coordinators come respectively from the fields of architectural design, landscape design, urban planning, and real estate appraisal. Additionally, the methodological approach and the team's configuration highlight the role of complementary disciplines that, through multidisciplinary exchange, are crucial to provide the necessary expertise to deal with complex territories. Indeed, economics, engineering, geomatics, sociology, statistics, and geography

are just some of the complementary competences that the group of 31 young researchers displays. Transdisciplinary research [46,47] is also addressed in B4R, with involvement of stakeholders and communities in co-design workshops and in co-visioning strategic scenarios [48].

The project targets the following specific outputs:

1.  An atlas as a result of the "Exploration" and "Co-design" phases, mapping places, actors, policies, and practices and collecting the proposal of small design interventions (operative branding actions) to accommodate social innovation initiatives, as accelerators of community resilience.
2.  The prototype of a web-based collaborative incremental platform (see Section 4.4) to support the elaboration of experience-based tourist itineraries and collect customers' data from the territories as well as to foster the interaction between users, tourist operators, public administrations, associations, and local stakeholders.
3.  A roadmap of strategic guidelines as an outcome of the co-visioning phase accompanying administrations in the formulation of supporting policies.

At the beginning of the research project, each research unit (RU) has selected a focus area (FA) to be investigated and compared in order to provide a significant sample of fragile inner territories and sketch a portrait of the current Italian situation. The four FAs are located in the Italian regions of the four RUs, Marche, Sicily, Trentino, Piedmont, and represent two Alpine and two Mediterranean regions.

Finally, focusing on a selected focus area (FA), each RU aims to complement existing programs, to foster new polycentric settlement models, and to boost the re-settling process with advanced design and strategic tools, working on territorial characterization, heritage, existing buildings, and natural resources. Starting from relational and experiential tourism, the process should lead to the enhancement and territorial development, and in general to activate new metabolisms for the analyzed marginal areas. In the following paragraph we describe the methodological approach of the "Exploration" phase, the focus of this article, specifying also the selection criteria for each FA.

## 3. The Methodological Approach

### 3.1. The Focus Areas Selection Criteria

The first phase of the methodological approach is the identification of the inner territories to be enhanced and reactivated. As outlined above, the National Strategy for Inner Areas (SNAI) defines as inner areas those areas that are significantly distant from the supply centers of essential services (education, health, and mobility), but rich in important environmental and cultural resources. The B4R approach goes beyond and considers not only the inner areas recognized by SNAI, but also other territories characterized by fragile socio-economic contexts.

Therefore, the B4R selection criteria focus on those territories with a population under 8000 inhabitants characterized by decreasing local supply of public and private services, population decline jointly with increase of aging population, progressive abandonment and degradation of the cultural and landscape heritage, and increasing economic stagnation. Moreover, B4R focuses on marginal areas whose touristic trends are decreasing or not constant throughout the year and thus can be improved by the development of minimal tourist infrastructures. The B4R targets are fragile inner territories that need to be enhanced by attracting a new population and touristic flows and by creating new employment circuits and new opportunities for local development. In this selection phase, different territorial contexts have been considered (Mediterranean, mountain, countryside, etc.): the proposed methodological approach is flexible and can be easily applied to different typologies of marginal areas. Coastal cities and urban and metropolitan constellations are usually excluded as they often deal with different dynamics and trends.

### 3.2. Exploration

The "Exploration" phase focuses on the investigation of the inner territories with the goal to analyze tangible and intangible features, qualities, and risks of each context. The methodological approach has been shared by all RUs, which worked in strong coordination to integrate quantitative and qualitative tools and methods in parallel. This approach led to the results explained in paragraph 4. The B4R methodological innovation lies in the outline of a non-linear and iterative process using the tools' combination as explained in Figure 1. The methodological approach can be used as a guide towards the creation of comparable results for the analyzed territories. The integrated tools are outlined as follows:

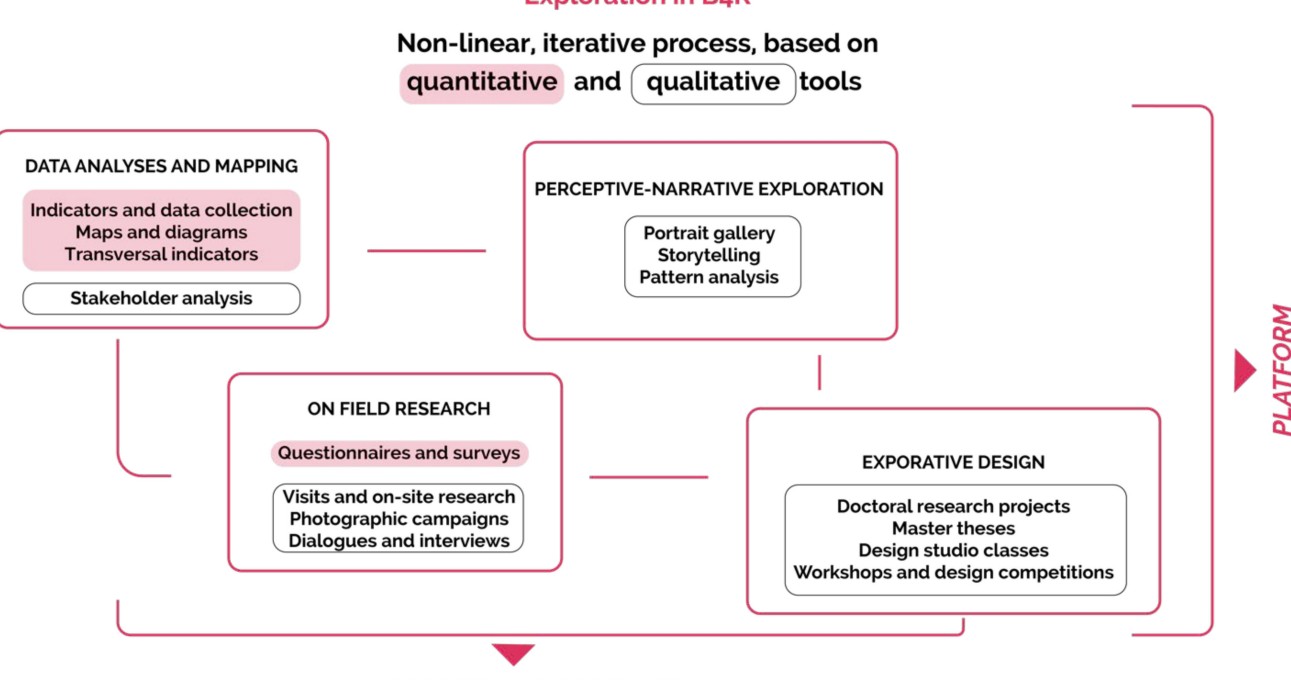

**Figure 1.** Methodological approach for the "Exploration" phase in B4R. Quantitative and qualitative tools are used in combination in a non-linear iterative process. Credits: elaboration M. Ferretti ©Branding4Resilience, 2020–2023.

1. Data analyses and mapping [49,50], including
   a. indicators and data collection from literature and databases,
   b. maps and diagrams based on a Geographical Information System (GIS),
   c. transversal indicators,
   d. stakeholder analysis (see Section 4.3) [51].

2. Field research, including
   a. visits and on-site research,
   b. photographic surveys,
   c. dialogues and interviews with local actors and communities [52].

3. Perceptive and narrative exploration [53,54], including
   a. portrait gallery,
   b. storytelling [55–57],
   c. pattern analysis [58].

4. Explorative design [40], including
   a. doctoral research projects,

  b.  master theses,

  c.  design studio classes,

  d.  workshops and design competitions.

The "Exploration", whose work package has been led by the Politecnico di Torino, is structured according to a general framework and four explorative dimensions, each covering specific themes to highlight relevant trends in the four selected FAs.

- Dimension 0: "Framework" explores basic coordinates to navigate in the analyzed context with a collection of preliminary information at regional and local scale to compare the different areas.
- Dimension 1: "Infrastructure, landscape and ecosystems" explores natural and landscape heritage, environmental risks, infrastructural networks, and connections.
- Dimension 2: "Built and cultural heritage, settlement dynamics" investigates the material and immaterial heritage, and it includes the analysis of the transformations of settlements and of cultural places and activities.
- Dimension 3: "Economies and values" analyzes the dynamism of the various productive sectors (industry, commerce and handicraft industry, agriculture, third sector), of the building industry and of the real estate market.
- Dimension 4: "Networks and services, community and governance models" examines models of governance and local development, planning tools and dynamism of the administrative sector, as well as forms of social and digital innovation.

These dimensions of exploration inform the data analyses and mapping, as well as other tools of the methodological approach. In particular, each dimension is structured in a variable number of sub-dimensions, which include one transversal indicator, one main map and some secondary maps, diagrams, and graphs representing relevant data for the analyzed topics. The transversal indicators serve to provide comparisons among different dimensions and sub-dimensions, suggesting cross-cutting aspects of the analysis. Main maps and transversal indicators have been discussed and coordinated among the four Rus and have been defined on the basis of their significance and as elements of comparison among the different territories.

All data analyses and mapping are based on literature review and on open data databases as well as on *ad hoc*-created new databases derived from data sources available at the national, regional, or municipal level (e.g., ISTAT—https://www.istat.it accessed on 6 August 2022, regional topographic maps and several others). This important knowledge base needs to be structured in a GIS, which organizes and stores harmonized, standardized, and georeferenced data [49,50]. The stakeholder analysis also constitutes a significant part of the "Exploration" phase and can be performed using a methodology based on four steps: identification of the main potential actors; scope definition; evaluation of the power and interest of each stakeholder; analysis of the results.

Furthermore, the "Exploration" phase includes other quantitative and qualitative tools, which are organized in three additional blocks (Figure 1). Field research is based both on quantitative tools, such as questionnaires and surveys with local community, and on qualitative tools, such as site-visits and photographic campaigns. These tools are used expand direct knowledge and experience of the places and to make contacts and perform interviews both with institutional actors and citizens. The interviews are conducted with a semi-structured set of questions and open dialogues [52]. Perceptive-narrative exploration is carried out with different forms of narration that aim to unveil hidden qualities of the analyzed areas. For example, four diverse looks at the territorial contexts are carried out with the galleries of photographs taken by four professional photographers that depict specific portraits of the areas, imprinting their perspective in the photographic work. With pattern analysis, the relational system of the territory is uncovered following Alexander's theory and applying it to the special analyzed context [40,58]. Storytelling is also employed both as an analytical and communicative tool, to better understand the place in a simpler and more direct way, and as a design tool to prefigure transformative visions [55–57].

Finally, to explore the focus areas, design is used as a "tool of reading, of conceptual innovation" which is useful for its selective and reassembling processes, as well as for its anticipatory and visionary quality [59]. To this end, explorative design [40] is used in different forms, involving students and young researchers that are supervised by research coordinators and guided towards the production of results and findings that, especially with doctoral research projects, are crucial for the advancement of knowledge and innovation in and for the areas.

The integrated application of quantitative and qualitative tools leads to the creation of one or more territorial portraits (see Section 4.5) that synthetically represent the values and risks embedded in the analyzed territories, as well as the transformative potential towards possible future development paths for the communities.

In the next paragraphs we share and discuss the results of the "Exploration" phase and the design of the collaborative platform, particularly focusing on the results of data analyses and mapping and on the synthesis of the overall explorative quantitative and qualitative analyses with the four territorial portraits.

## 4. Results

### 4.1. The Selected Focus Areas

The selected focus areas (FA) of the project are four fragile inner territories (two Mediterranean regions and two Alpine areas), generally characterized by aging population, progressive abandonment, economic stagnation, difficult accessibility, and access to primary services. For each FA, the RU assumed a specific theme and selected a specific pilot case (PC) within the area.

The Università Politecnica delle Marche (UNIVPM) assumed the "Built Heritage" theme and focused on the pre-Apennine Mediterranean area between Urbino and Fabriano (Marche region), characterized by the presence of small medieval villages and traditional rural settlements; important natural areas (Monte Catria, Monte Nerone, Regional Park Gola della Rossa) and water landscapes (Cesano and Metauro); renown regional food products (truffles, bread, wine, artisanal beer).

The Università degli Studi di Palermo (UNIPA) assumed the "Co-creative communities" theme and focused on settlement villages and communities in the Sicani Area (Southern Sicily), with a particular attention to some re-activation processes and neo-rural practices which are generating creativity for tourism and social innovation in the selected FA. For example, it is worth mentioning Cianciana, Sambuca di Sicilia, Caltabellotta, Sant'Angelo Muxaro, which represent municipalities where communities are exploring different forms of relational tourism.

The Università degli Studi di Trento (UNITN) assumed the "Thermal Water" theme and focused on small thermal villages in the mountain region of Trentino Alto Adige, such as Rabbi, Peio, Ponte Arche, Vetriolo, and Levico Terme. A particular attention was paid to the value of the whole water system, by promoting the reuse and enhancement of elements that are part of their territorial capital, as well as to blue and green infrastructures, able to create healthier and greener living habitats and to activate urban and regional regeneration processes.

The Politecnico di Torino (POLITO) assumed the "Natural environment" theme and focused on hamlets and villages in the Alta Valsesia Mountain area (Northern Piedmont), characterized by the presence of a significant natural and cultural heritage. For example, in addition to the Sesia Val Grande Geopark and the Alta Valsesia and Alta Val Strona Natural Park, that is the highest park in Europe, there are numerous valleys where the wild environment surrounds ancient Walser settlements and abandoned hamlets that need to be enhanced and repopulated.

### 4.2. Exploration

During the "Exploration" phase not all RUs performed the exact articulation of all above-mentioned tools, but all have collaboratively envisaged and shared the common quantitative analysis (elaboration of maps, diagrams, and transversal indicators) based on a general framework (dimension 0) and four explorative dimensions, detailed in the following sub-sections.

#### 4.2.1. Dimension 0

The Dimension 0 "Framework" provides the basic coordinates to navigate in the analyzed context, outlining a series of maps and diagrams. The Dimension 0 can be articulated in 2 sub-dimensions. Description and contents are briefly illustrated in Table 1.

**Table 1.** The "Framework" sub-dimensions, ©Branding4Resilience, 2020–2023.

| Sub-Dimension | Content | "Transversal Indicator" |
|---|---|---|
| Regional Framework | Map of the whole region with highlight of the FA, regional, provincial and municipal boundaries, main and most significant infrastructural network for each FA (e.g., roads, highways, waterways), municipal centers (SNAI 2014–2020), intermunicipal centers (SNAI 2014–2020), existing settlements. | Surface in $Km^2$, population density in comparison with the regional indicator, and land consumption compared to regional average. |
| Focus Area Framework | Map of the whole focus area, regional, provincial and municipal boundaries, main and most significant infrastructural network for the FA (e.g., roads, highways, waterways), existing settlements, map of all municipalities, with indication of areas, residents, density, average age of the population, aging index, composition of the population according to age categories. | Number of municipalities in the FA, overall number of the FA (2019), and population trend for the whole FA (2010–2019). |

The Dimension 0 highlighted some important numbers, for example that the B4R FAs involve 57 municipalities where 105612 inhabitants reside and where the average population density is 31.7 inh/$Km^2$. Many differences are noticeable in the four areas in terms of average density of the population (56.68 inh/$Km^2$ in the Sicani FA, 25.51 inh/$Km^2$ in the Val di Sole FA, 39.79 inh/$Km^2$ in the Appennino Basso Pesarese Anconetano FA, 6.69 inh/$Km^2$ in the Val Sesia FA) and also in terms of size and absolute number of inhabitants, the largest being the Sicani FA with 1191.40 $Km^2$, 18 municipalities, and 54025 inhabitants, the smallest being the Val Sesia with 525.35 $Km^2$, and 3515 inhabitants, with a high fragmentation of the municipalities (17 in total).

Despite the diversity of many indicators, the collection of preliminary information was useful to identify the areas and start a comparative exploration of the different territories.

#### 4.2.2. Dimension 1

The Dimension 1 "Infrastructure, landscape and ecosystems" explores the main aspects related to the natural capital combined with the physical network systems in the FAs. It highlights the interlinked impacts on the landscape transformations and territorial structures among natural and biodiversity features due to infrastructure and anthropogenic changes. This dimension also investigates the state of development of technological and telecommunication networks to understand the impact of digital divide. These systems need to be understood in relation to the most updated numbers of accommodations, public facilities (such as hospitals, schools, and medical services), and the population age. The contents of each sub-dimensions are briefly illustrated in Table 2.

**Table 2.** The "Infrastructure, landscape and ecosystems" sub-dimensions, ©Branding4Resilience, 2020–2023.

| Sub-Dimension | Content | "Transversal Indicator" |
|---|---|---|
| Natural and landscape heritage | Natural protected areas, natural parks, UNESCO natural sites; water sources; terraced landscapes; roman pathway; archeological sites; panoramic viewpoint; monumental trees; environmental point of interests; valued landscapes (*paesaggi di pregio*). | Natural areas—woods, agricultural, and protected areas (Km$^2$). |
| Geomorphological and hydrographic character | Land use (Corine Land Cover 2018); hydrographical networks and systems; hydrographical protected areas; areas of paleontological, mineralogical, and stratigraphic interest; historic quarry; rocky areas. | Climatic values of temperatures (°C) and precipitations at regional scale (2014). |
| Natural and anthropogenic risks | Hydrological hazards (flooding, avalanche, landslide, storm); anthropic hazards (e.g., industry, agriculture); hazards from physical agents (e.g., noise, electromagnetic camp). | Risks percentage (high, medium, low); numbers of extreme events in the last 10 years. |
| Infrastructural networks and mobility | Infrastructural network for fast mobility (roads, railways, airports, train stations, bus stations, parking lots, e-charge stations, sharing mobility); infrastructural network for slow mobility (bike paths, trails, mountain routes). | Number of accommodation facilities; number of schools; number of public services (hospitals, clinics, medical services). |
| Technological infrastructure and telecommunications networks | Internet coverage; primary urbanization (water, gas, electricity, sewerage); digital divide. | Variation of population by age groups (0–14, 15–64, 65+) of aging population in percentage (2010–2019); density of families served by ADSL technology (2018). |

Dimension 1 identified the presence of high natural capital in all territories, but at the same time high hydrogeological, seismic, and anthropogenic risks. As an example, the investigation of the Dimension 1 for the Val di Sole FA in Trentino stressed the presence of a high natural capital (e.g., protected areas, natural parks, ecological corridors) with a relatively limited spread of settlements. Thereby, nearly half of the territory is subject to protection and preservation measures (70% of the 611.53 Km$^2$ total surface of the FA are covered by woods). Instead, the built heritage is often underused or abandoned. The geomorphological and hydrographic features highlight the large characterization of the territory based on the water system intended as the most valuable and vital system (Figure 2). A variety of forms of water emerged from the explorative maps: rivers, lakes, glaciers, springs, and thermal waters generate the most precious resources as well as the cultural values in all the FAs. The natural and anthropic risks show a fragile land characterized by a medium and high hydrogeologic risks (flooding, avalanche, landslide, storm). The infrastructure networks and mobility systems identify only one main route that connects the valley to the nearby regions with a weak public transport system. In particular, the two valleys of Rabbi and Peio are "dead-end" systems for vehicles which worsens their marginality. This condition is recurring in the other Alpine FA in Piedmont, while in the FA in Sicily, the poor state of preservation and maintenance of the mobility infrastructure networks increases the accessibility difficulties to these territories and becomes a leverage for their abandonment. Also, in the Marche region the distance from main infrastructural hubs such as the international airport of Ancona and its harbor testifies to the poor accessibility of the FA, a condition which represents a potential leverage to its depopulation. In all the FAs, the lack of organized and efficient public transport increases the convenience of private mobility usage both for timesaving and comfort. However, a widespread pathway towards slow mobility (bike paths, trails, mountain routes) is noticeable in all territories, offering a unique opportunity to accommodate and promote sustainable tourism flows and dynamics. In the Val di Sole FA the telecommunications networks are well structured and only 5% of the population is not reached by broadband (2013). However, the other FAs

show a problematic lack of connections that significantly increment the digital divide and marginality of the territories.

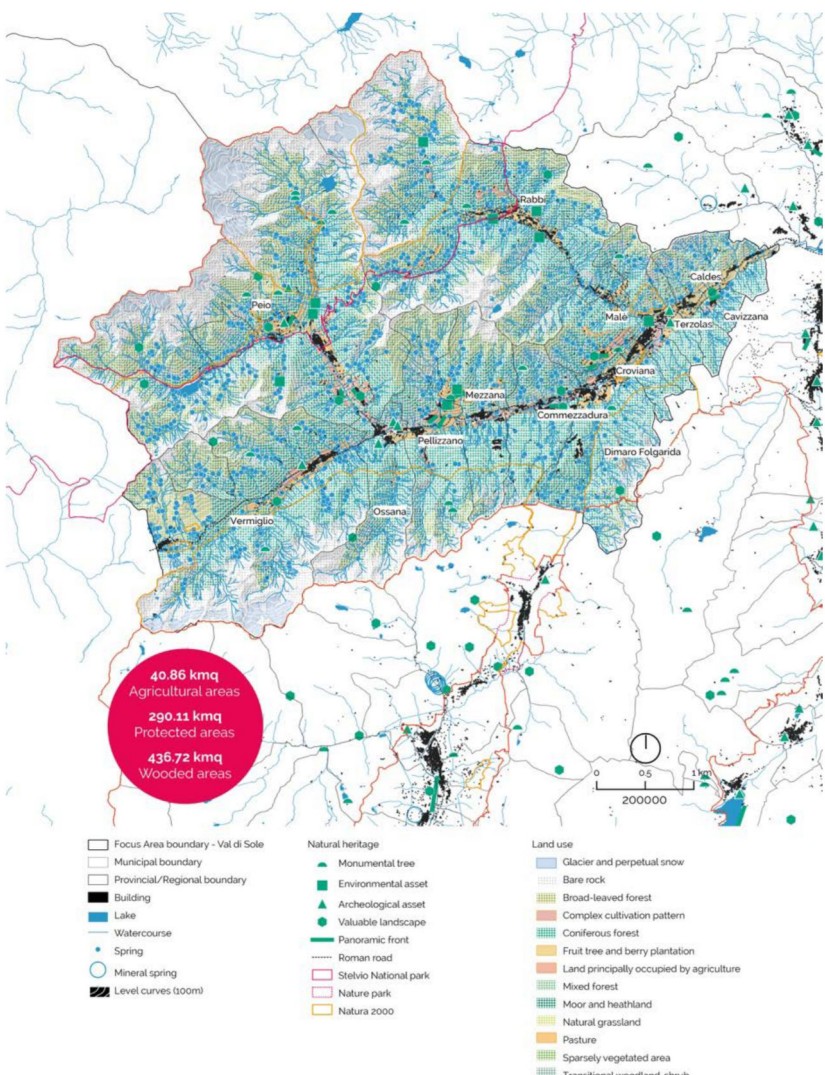

**Figure 2.** The Val di Sole Focus Area. "Natural and landscape heritage" map. Credits: coordination S. Favargiotti, elaboration by M. Pasquali and C. Chioni, 2021, ©Branding4Resilience—UNITN, 2020–2023.

### 4.2.3. Dimension 2

Dimension 2, "Built and cultural heritage, settlement dynamics", explores the main aspects related to the rich material and immaterial heritage of the FAs and studies the structure and the dynamics of transformation over years of buildings and settlements. This dimension also investigates tourist presence and flows in relation to the cultural assets of the area, together with the analysis of number and types of accommodations, and of cultural and leisure activities. Most notably, the analysis was focused on detecting the cultural landscape values but also the use of built heritage, the current activities, the state of conservation, the level of protection and constraints, and the type of property. At a territorial and urban scale, the type of settlement and the historical phases of its development were examined, as well as the land consumption data in relation to the demographic trends. The contents of the dimension are displayed in Table 3.

**Table 3.** The "Built and cultural heritage, settlement dynamics" sub-dimensions, ©Branding4Resilience, 2020–2023.

| Sub-Dimension | Content | "Transversal Indicator" |
|---|---|---|
| Built heritage and resources: functions and use, protection, and conservation | Buildings, primary and secondary buildings (residential use/other use), in use/unused buildings (in percentage); conservation rate, protected and listed buildings, confiscated buildings, state land or building, type of property. | Variation of aging population in percentage (2010–2019). Percentage of abandoned buildings (2011). |
| Structure and transformative dynamics of settlements | Settlement typology and size, population density for municipality based on seasonal change, population variation based on age categories, birth-death rates and population trends indexes, variation of the composition of families, percentage of abandonment of settlements, historical evolution of anthropic settlements according to main reference dates (end of XIX cent., around 1920s, after the 2nd WW, end of XX cent.). | Variation of resident population (2010–2019). Variation of land consumption in percentage (2010–2019) |
| Tourist flows and accommodation facilities | Restaurants and food structures, protected and listed buildings, tourist pressure or intensity, accommodation capacity (number of beds per 1000 inh.), touristic density, index of average stay length, arrivals trends, presence trend. | Average permanence in the FA compared to regional average (2019). |
| Networks, places, and cultural activities | Cultural networks and events, museums, libraries, archives, schools and educational buildings, cultural associations and community centers, tourist density, tourist presence, tourist itineraries linked to landscape, enogastronomy, culture, religion, cultural centers, UNESCO sites, religious communities boundaries, cultural communities boundaries, "authentic Italian villages" and "*borghi più belli d'Italia*", classification of villages according to artisanal and artistic production, type of enogastronomic products and food excellence, networks and local associations, cultural vibrancy, trends of birth and death of cultural places. | Variation of resident population according to seasons (2019). |
| Sports and leisure offerings | Thermal and wellbeing centers, number and typology of sport and leisure activities, sport equipped areas, parks and protected natural areas, hiking paths, ski compounds, beach resorts, sport events of local and national level, sport experiences, sport services and buildings. | Ratio of parks area to inhabitants. Ratio of sport facilities to inhabitants. |

As an example, the investigation of the Dimension 2 for the Appennino Basso Pesarese e Anconetano FA (Figures 3 and 4) led to focus on the presence of several protected buildings (479 listed buildings) which are mainly located in the area's historical centers, even if there is a quite high diffusion of protected built heritage in rural areas and in spread settlements, especially in the municipalities of Sassoferrato and Arcevia. The diffused presence of high value buildings throughout the analyzed territory, though, does not match with the scarce conservation levels of a significant part of this heritage, particularly to the north of the area, in the province of Pesaro-Urbino. Most building stock is destined for residential use and only a low percentage is dedicated to other types of use, such as commerce, industry, services and accommodating structures, culture, and sport. Some important data are the relatively large presence of abandoned buildings (684 in 2011), which is coupled by an increasing percentage of the population over 65 years old (+3.7% in the period 2011–2019). In parallel, the study of settlements' structure and their urban development over centuries was useful to confirm the qualities of this particular type of cultural landscape in central Italy, which is characterized by a strong relationship between settlements and geomorphological and landscape conditions. Villages and towns, that in this Apennine region typically date back to medieval times, have been built according to topographic conditions, usually on hilltops, or along river valleys, to gain natural defense from possible outside attacks. More recent development patterns are noticeable towards the coastline, in valleys and planes, to exploit more favorable conditions for agricultural or industrial use. Typical central-Italy historical villages are often known as *borghi*, some of

which are listed in special protection and touristic networks, such as the "*Borghi più belli d'Italia*" for Sassoferrato. While the condition of isolation of these settlements has to some extent contributed to the conservation of their qualities, the lack of linkages and connections highlighted in the "Exploration" stressed the need to act towards a major integration in cross-cutting networks, be they cultural, political, or governance-related ones.

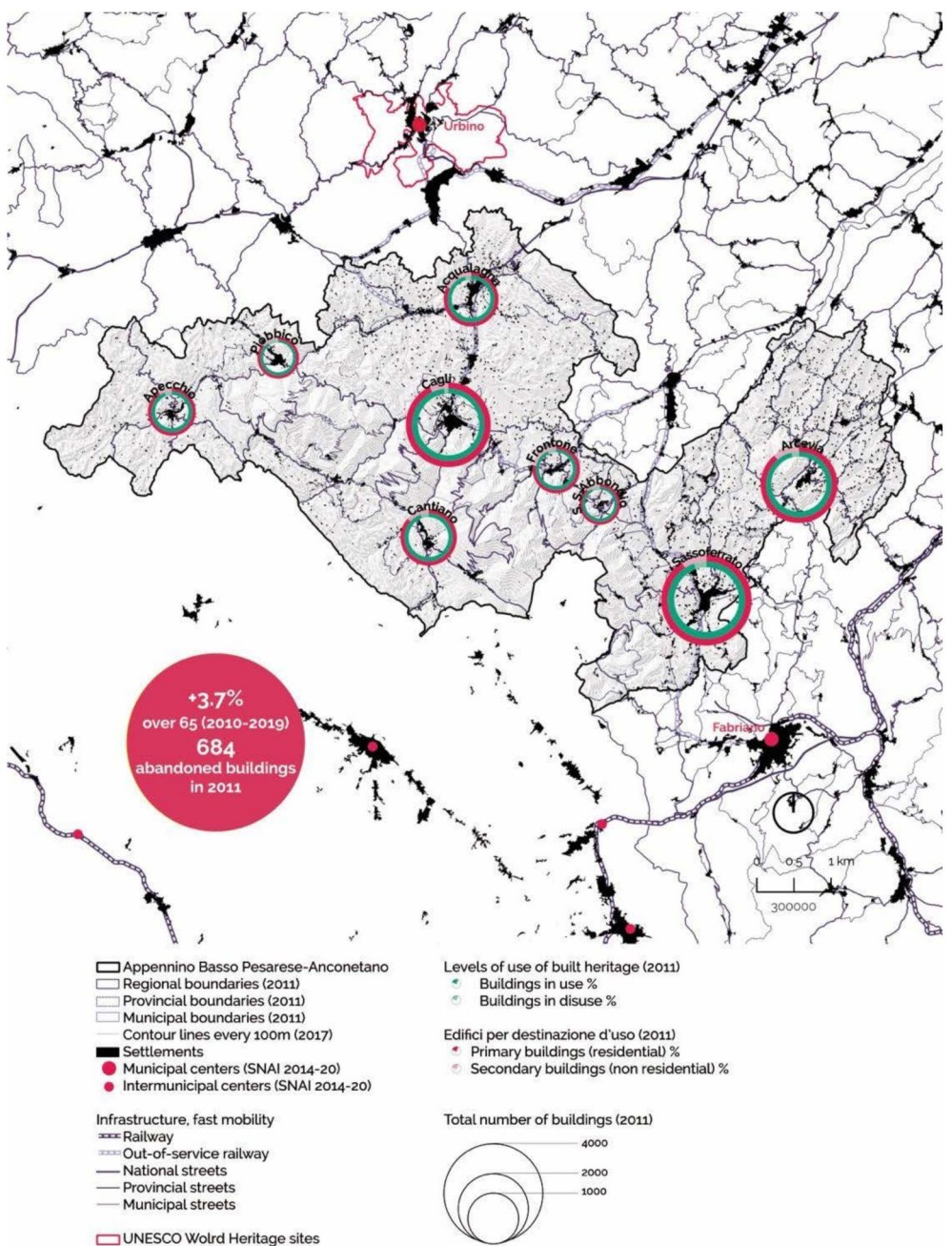

**Figure 3.** The Appennino Basso Pesarese e Anconetano Focus Area. "Built heritage and resources" map. Credits: coordination M. Ferretti, elaboration by M.G. Di Baldassarre and C. Rigo, 2021, ©Branding4Resilience.—UNIVPM, 2020–2023.

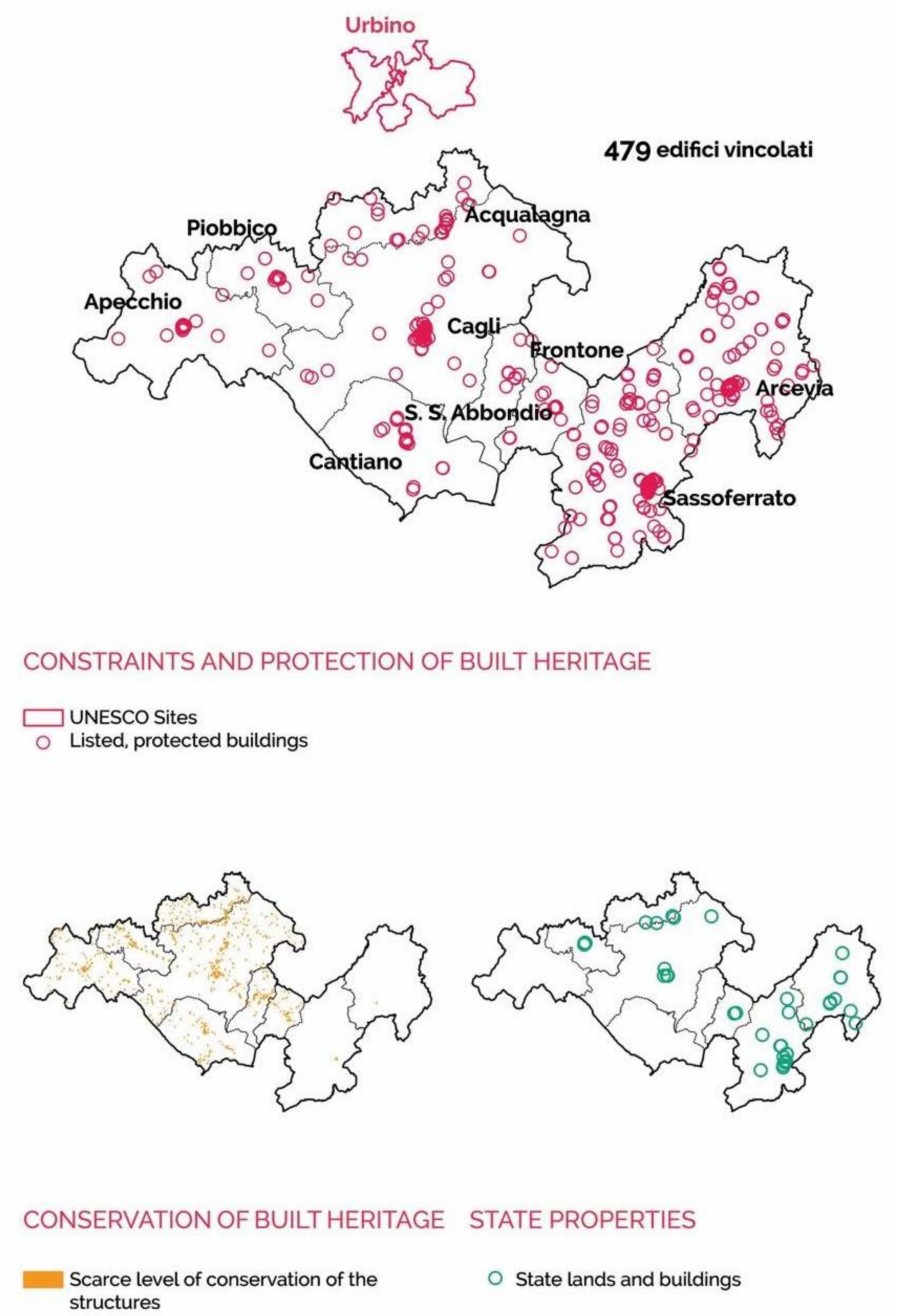

**Figure 4.** Diagrams on built heritage for the Appennino Basso Pesarese e Anconetano Focus Area. Credits: coordination M. Ferretti, elaboration by M.G. Di Baldassarre and C. Rigo, 2021, ©Branding4Resilience—UNI.VPM, 2020–2023.

### 4.2.4. Dimension 3

The Dimension 3 "Economies and values" explores the main aspects related to economic context of the analyzed area. It investigates the dynamics of the primary, secondary, and tertiary sectors, in order to highlight the dynamism of different companies by considering the variation trends in the last years. Particular attention is paid to the building sector, whose economic dynamism is analyzed by studying the real estate market and the construction activity trends. Dimension 3 can be articulated in 5 sub-dimensions, whose description and contents are briefly illustrated in Table 4.

**Table 4.** The "Economies and values" sub-dimensions, ©Branding4Resilience, 2020–2023.

| Sub-Dimension | Content | "Transversal Indicator" |
|---|---|---|
| Dynamism of the primary sector | Analysis of the density of active farms, youth-run businesses, female-run businesses, variation of the density of farms in the last 10 years, birth and mortality rate of firms in the primary sector. | Entrepreneurial vibrancy in the primary sector (2010–2020). |
| Dynamism of the secondary sector | Analysis of the density of active enterprises, youth-run businesses, female-run businesses, variation of the density of enterprises in the last 10 years, birth and mortality rate of firms in the secondary sector. | Entrepreneurial vibrancy in the secondary sector (2010–2020). |
| Dynamism of the tertiary sector | Analysis of the density of active service industries, youth-run businesses, female-run businesses, variation of the density of service industries in the last 10 years, birth and mortality rate of firms in the tertiary sector. | Entrepreneurial vibrancy in the tertiary sector (2010–2020). |
| Dynamism of the real estate market | Analysis of the main real estate market trends, in terms of listing prices, transaction prices and rental prices variation rates, number of property listings on the market, number of real estate transactions and their variation in the last 5 years. | Mean listing price variation rate (2010–2020). |
| Construction activity and sustainability | Analysis of the construction activities carried out in the last 10 years, with a particular attention to the land consumption and other sustainability aspects such as the achieved energy performance levels and cadastral categories. | Percentage of buildings built before 1971. |

For instance, the analyses related to the "Dynamism of the tertiary sector" sub-dimension highlighted different issues and opportunities in the Alta Valsesia FA (Figure 5). Although the general trend of the dynamism of the tertiary sector is positive (+11.50% from 2010 to 2020), the territorial distribution highlights some discrepancies. In fact, the map in Figure 5 shows that the economy of several municipalities is negatively impacted by commercial desertification and that this phenomenon is mainly affecting the lateral valleys (Val Sermenza and Val Mastallone). Commercial activities, restaurants, and accommodation, such as hotels, bed and breakfasts, and farmhouses, are mostly located in two municipalities (Alagna and Scopello) as well as along the road that runs through the main valley (Val Grande), connecting Varallo to Alagna. In the smaller villages the number of bars and shops that are permanently closed is gradually increasing, not only due to the COVID-19 pandemic, but also to the increase of the aging population and the absence of young people interested in working there and in investing in new local businesses.

The importance of the tertiary sector activities goes beyond the economic impact on the territory since it has also a very important social relevance, above all if the analysis is focused on food stores, which also represent points of social aggregation. In the analyzed area they are very few, above all when compared to the non-food stores. The presence of weekly street markets is emblematic in Figure 5: there are 2 medium-size markets (31–100 market stalls) in Alagna and Scopello, while in only other 3 municipalities are there small-size markets (no more than 5 market stalls).

4.2.5. Dimension 4

Dimension 4, "Networks and Services, Community and Governance Models", explores planning tools, governance models, networks, and the policy framework in which the local governments of the four analyzed FAs operate, and which determine their relationship with key policy actors and with community and interest groups. The dimension also explores the offer of local and supra-local services, the community and its fragility, the social vibrancy, and some ongoing innovation trends.

Dimension 4 is articulated in 9 main subdimensions that include maps, schemes, and graphics, as well as transversal indicators, as shown in Table 5.

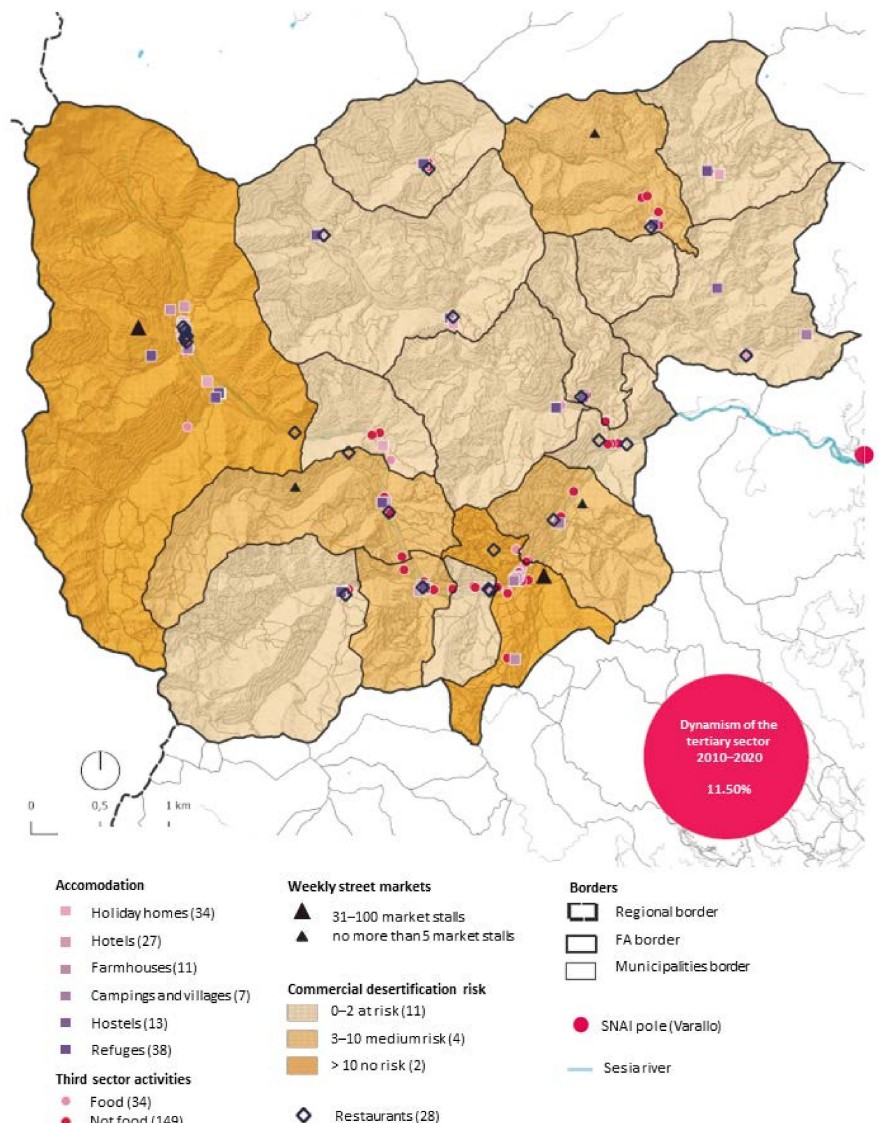

**Figure 5.** The Alta Valsesia Focus Area "Dynamism of the tertiary sector" map. Credits: coordination D. Rolando, elaboration by A. Barreca and G. Malavasi, 2021, ©Branding4Resilience.—POLITO, 2020–2023.

**Table 5.** The "Networks and Services, Community and Governance Models" sub-dimensions, ©Branding4Resilience, 2020–2023.

| Sub-Dimension | Content | "Transversal Indicator" |
|---|---|---|
| Planning Tools | The map describes local plans on the base of the year of adoption and municipal expenditure on spatial planning and construction. In addition, detected metropolitan or provincial plans based on the specific competences of administrations. | Total spending on spatial planning and construction. |
| Planning Dynamism | In addition to the funds used by the different municipalities in the programming period 2014–2020, the map describes how many municipalities are engaged with supra-local networks such as local action groups funded by Leader+ program or SNAI. It is relevant in terms of capacity building and opportunity in future development. | Average age of administrators in FA, checking whether local project dynamism is related to the age of administrators. |

**Table 5.** *Cont.*

| Sub-Dimension | Content | "Transversal Indicator" |
|---|---|---|
| Administrative vitality | The map describes the amount and areas of intervention of the funds used by the municipalities of the AF within the 2014–2020 programming. | Total project costs 2014–2020 programming in the FA. |
| Level of Peripherality | The map describes the peripherical status, according to National Department for Development and Territorial Cohesion in the SNAI framework, in addition to the availability of basic services such as schools and health services. | % of population reached and not reached by fixed-line broadband (Asymmetric Digital Subscriber Line—ADSL—FTTC (FTTS)—FTTH) below or not below 100 mbps (effective capacity). |
| Type of Services | The map describes the availability of basic services such as schools and health services in addition to other local services (as sport and culture) and supra-local services. | Average expenditure per capita in the programming period 2014–2020. |
| Effectiveness of public spending versus available services | The map describes the public spending in relation to the type and number of services available. | Per capita taxable income. |
| Community and Fragility | The map describes the municipal average per capita income in addition to the presence of fragile population (>14 years old and <65 years old). | % of population fragile population in the FAs. The maps show a high percentage of the population that is fragile, especially in the elderly component (>65). |
| Communities, social media and web marketing | The map describes the municipal in terms of social vibrancy of official social media profiles of the municipalities. | Number of supra-local web communities. |
| Innovation Experiences | The map describes the main associations present in the territory and relevant and significant regeneration processes, repopulation, and valorization experiences to describe ongoing innovation trends, where present. | % of population 15–34 years old representing the potential most active range of population for innovation processes. |

The community and fragility map for the Sicani FA (Figure 6) describes the area from the point of view of the fragility of the settled community, taking into account elements that highlight critical factors in terms of demography (population under 14 and over 65 years of age, old-age index, population distribution by age group and population density, working population, and structural dependency ratio) and economy (per capita income, employment rate).

The population under the age of 14 and over the age of 65 (in the Sicilian FA 40% of the FA population) describes the presence of those population groups that are more fragile and more in need of care and assistance services. For this reason, the map also describes the presence of school services or health facilities with DEA centers (Emergency Health Departments), which in the Sicilian FA are located exclusively in the cities of Agrigento, Sciacca, and Castelvetrano, i.e., distant from the municipalities of the FA and served by poor road infrastructures. As described in the map, in the Sicani area the highest values of fragile population are recorded in the municipality of Santa Margherita Belice, followed by Sambuca di Sicilia, and Bisacquino and the value of per capita income is an average in the FA of EUR 12,469 compared to the regional average value that in 2019 was EUR 15,846, already significantly lower than the national average. Looking at the trend in the Sicani FA, the municipality with the highest per capita income is Palazzo Adriano with EUR 14,091 followed by Bivona with EUR 14,040. The average value of the old-age index (2020) is 266.46 compared to the regional average of 159. The structural dependency index, which represents the social and economic weight of the inactive population (0–14 years and over 64 years) on the active population (15–64 years) also has an average value in FA of 62.72% compared to the regional average of 54.40%. With reference to the employment rate (2018)

there is an average in the FA of 47.5% in 2018 compared to the regional average of 44.1%: the highest value is that of Villafranca Sicula with 59.2% and the lowest is that of Cianciana with 40%.

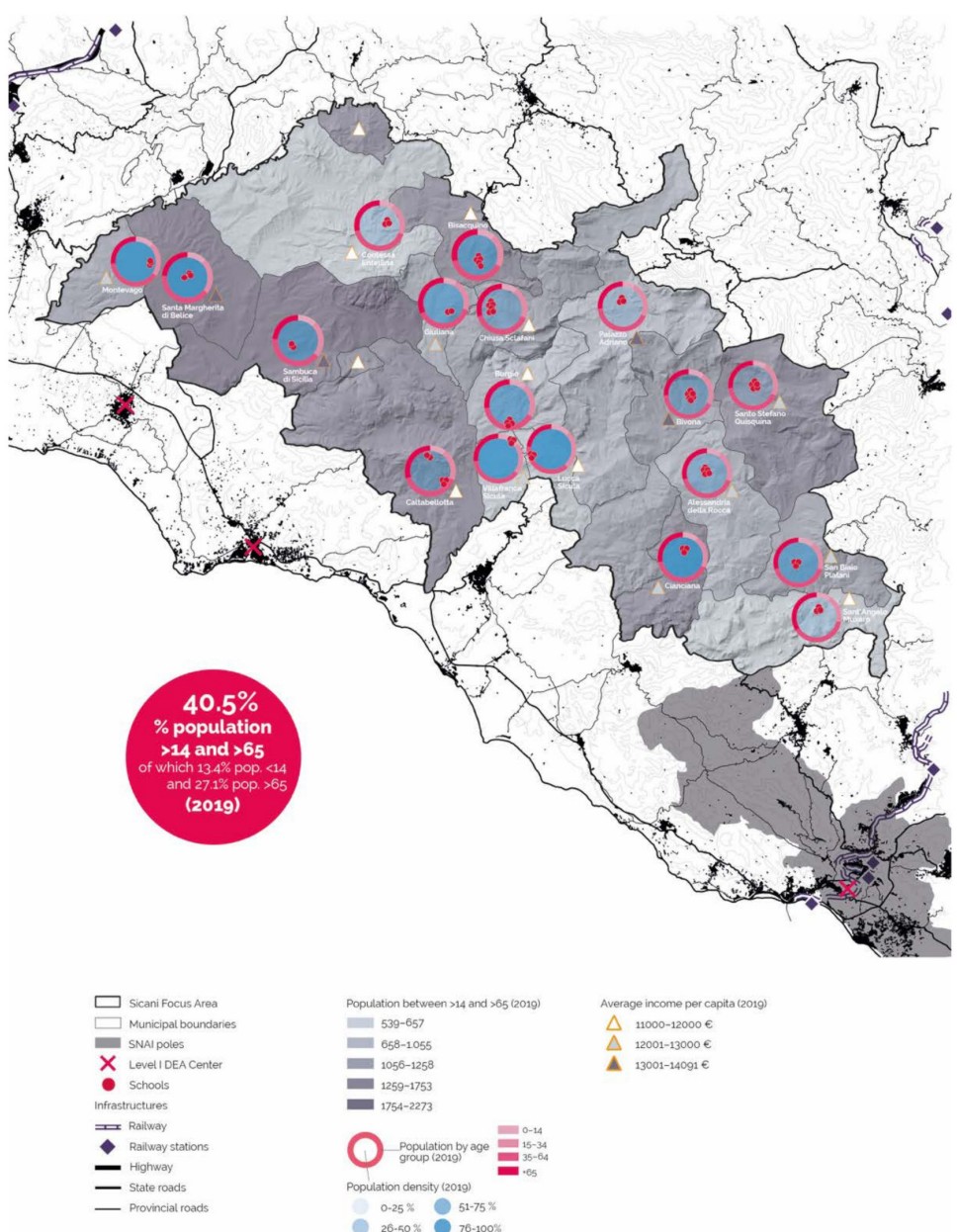

**Figure 6.** The Sicani focus area community and fragility map. Credits: coordination B. Lino, elaboration by B. Lino and A. Contato, 2021, ©Branding4Resilience—UNIPA, 2020–2023.

### 4.3. Stakeholder Analysis and Interaction with Local Actors

Parallel to data analyses and mapping, the proposed methodological approach suggests carrying out a stakeholder analysis in order to identify the key actors to be involved in possible enhancement processes in the focus area. To this aim, the four RUs of the B4R project followed the above-mentioned methodology based on 4 main phases. According to the first phase (1. Identification of the main potential actors), each RU listed the main public and private subjects in the analyzed FA and split them into two categories: "local stakeholders" (municipal level) and "territorial stakeholders" (inter-municipal, regional, and national levels). Subsequently, the RUs jointly defined the main scopes of the analysis (2. Scope definition), according to the B4R research objectives and to the four dimensions

that were also used to structure the "Exploration" analyses. Figure 7 shows the 8 challenges derived from a debate among the RU regarding the scope definition.

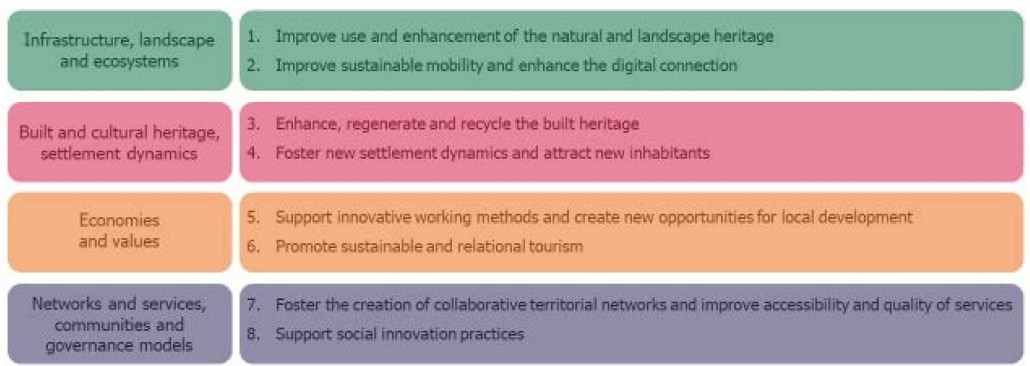

**Figure 7.** The 8 challenges assumed for the stakeholder analysis, ©Branding4Resilience, 2020–2023.

Assuming the scope of each challenge, each RU evaluated the power and the interest of the previously identified stakeholders (3. Evaluation of the power and interest of each stakeholder) by using the power-interest matrix [51], which supports the identification of 4 stakeholder typologies: the key stakeholders that have to be closely managed (high power, high interest); the marginal stakeholders that should be monitored (low power, low interest); the stakeholders that need to be kept satisfied (high power, low interest) and those that need to be kept informed (high interest, low power).

The resulting 8 matrixes (one for each challenge) supported the analysis of the relational dynamics between stakeholders and the territory (4. Analysis of the results) and the identification of those main players with an active role in the territory enhancement processes, to be necessarily interviewed and involved in the next phases of the B4R project. The analyses carried out in the four FAs led to the identification of different key stakeholders, which were involved during the planning and development of the co-design workshops, the second phase of the project.

For example, in Alta Valsesia the Unione Montana dei Comuni della Valsesia and the "Terre del Sesia" local action group (LAG) emerged as the most strategic stakeholders, jointly with each administrator (mayor) of the municipalities located in the FA. The Unione Montana is a second level body that represents the whole territory of Valsesia (30 municipalities), manages numerous territorial services, and pursues a unitary strategy for the development of the whole area by means of the long-term plan for the territory's socio-economic development. Instead, the "Terre del Sesia" LAG is a consortium of both public and private bodies that supports rural development projects through the enhancement of local traditions and culture, typical Valsesian products, and traditional agricultural and craft activities of the territory.

The importance of the role of the municipalities located in the FA emerged from all the stakeholder analyses, but also some differences can be highlighted. In the Appennino Basso Pesarese e Anconetano, the following different stakeholders emerged from the analysis: the Mountain Union Catria and Nerone, leader of the SNAI Strategy, some change makers/social innovators, and the tourism operators "Happennines srl" for the paradigmatic case of Sassoferrato. The unifying role of the Mountain Union is relevant, especially for the implementation of common projects within the SNAI strategy, such as the Apennine Asylums, creative residencies implemented in all the nine municipalities of the FA. Instead, from the dialogues and interactions that followed the stakeholder analysis, it was clear that the associations and other private operators in the Marche FA, as well as some social innovators and change makers, have difficulties creating networks and interacting with local institutions. This aspect has been a crucial element especially in the design of the B4R platform, where a specific "Network" functionality/section has been designed right in response to these needs (see Section 4.4). In the Sicani area other stakeholders emerged: the

"SNAI Sicani" and "Sicani LAG" (intermediate actors), "Via delle Rondini", "Sikanamente", "Rifai" (territorial facilitators/local associations), and the tourism operator "Val di Kam". In the Sicani FA, the LAG (local action group) has, over the years, taken on a directing role in the processes of the emergence of a common identity in the area, also through effective participatory processes, in contrast to the SNAI Sicani, which, on the other hand, has had a less significant impact in terms of its ability to involve territorial actors. Finally, the "special" condition of the Autonomous Province of Trento offered the opportunity to confront additional stakeholders in Val di Sole: "Visit Val di Sole", the regional agency for tourism promotion, the Stelvio National Park, the thermal consortium, the *Comunità di valle* (lit. Valley Community), the Municipalities of Peio and Rabbi, the fluvial park of Nove river (*Rete di Riserve Alto Noce*), the local cultural associations and activators such as the Ecomuseo *Piccolo Mondo Alpino* in Peio and *Mulino Ruatti* in Rabbi, the ASUCs (lit. the "Separate Administrations of Civic Uses"), and the Mountaineers Society of Trento (SAT).

*4.4. Platform*

Running in parallel with the "Exploration" phase, an important result of B4R was the creation and implementation of a pilot incremental, collaborative, web-based platform for the branding of the analyzed territories. The platform guides visitors throughout the territory and, working through artificial intelligence, utilizes profiling to generate experience-based and highly customized itineraries for specific target users. The B4R platform aims to improve communication and offer triggering transformation, new job opportunities, and local development in the areas, and thus ultimately increase the resilience of communities living in marginal contexts. To reach these goals, the platform has been envisaged not as a supporting tool for a territorial marketing strategy, but with the specific aim to build a place for exchange and networking for tourists, communities, and administrations experiencing, living, and operating in the FAs. Following the Airbnb model, an important aspect of the collaborative platform is that users can simultaneously be consumers and providers.

The platform has been collaboratively designed with an interdisciplinary approach involving all project partners led by the computer scientists and engineers of the UNIVPM RU and its first pilot has been tested in a small town of the Marche Region FA. During the first two years of the project, achieved results for the B4R platform have been the following:

- analysis of best practices;
- collaborative "Discovery" phase to identify functional requirements and types of users and uses;
- definition of six hypothetical key users, namely economic and tourist operators, administrations, future residents, travelers, associations of citizens, temporary residents;
- architectural design of the pilot platform, organized in search engine, network, creation of itineraries, stories, dashboard and business intelligence;
- involvement of a local pilot case (PC) to collect real users' data and improve machine learning for user profiling.

One of the expected impacts of the platform is the capacity to reactivate marginal territories through people's experiences. The "stories" section will gather visitors' impressions and share the providers' stories of excellence. Together with other operative branding actions performed in the project, the B4R platform will thus contribute to build the branding and enhance the image of the territory from outside to attract new people.

*4.5. Territorial Portraits*

Results of the "Exploration" phase, the territorial portraits outlined guiding themes and development trends for each FA, addressing demographic, social, economic, spatial, settlement, architectural, natural, and infrastructural aspects. Collecting information and impressions through quantitative and qualitative analysis—data analyses, mapping, surveys, on field research, portrait gallery of photos, study of literature, perceptive-narrative analysis, pattern analysis, stakeholder analysis, interviews, explorative design, storytelling,

etc.—the complex logic and relational structure of the four FAs has been unfolded and interpreted in their composite articulation of values and risks. The combination of high value and high risk determines the areas that are more sensitive, and thus those areas where co-design and co-visioning processes are more needed. These processes are driven towards the preservation and enhancement of those values, all the while trying to minimize the natural and anthropic risks in order to envisage collective meaningful interventions to increase community resilience. The territorial portraits help focusing on these priorities.

4.5.1. Appennino Basso Pesarese E Anconetano, Marche Region: Re-Settling the Central Apennine Starting from Built Heritage, Networks, and Creativity

The Appennino Basso Pesarese e Anconetano is the Marche Region's pilot inner area for the SNAI. This pre-Apennine area to the north of the region includes nine municipalities with 32,037 residents (Istat 2019) and an average population density of 37.86 inhab./Km$^2$. Extended marginalization processes include a demographic decrease of $-9.33\%$ (2010-19), reduced accessibility, and lack of offered education and health services. The "Exploration" highlighted a polarized settlement structure with heritage medieval villages and traditional rural settlements, important natural areas and water landscapes, renown regional food products, numerous protected and listed buildings, and a general diffuse quality of the historical centers and architectures. Weaknesses regard also seismic and hydrogeological risks, aging population, progressive abandonment, economic stagnation. A striking data, especially in the light of the current debate on the digital divide of inner territories, is the fact that 52.8% of inhabitants are not reached by broadband (source: AGCOM 2020). These conditions also impact the territorial presidium and care of the area, producing the abandonment of built heritage (684 abandoned buildings in 2011) and of the industrial or proto-industrial legacy (e.g., windmills, concrete factories, agricultural consortia), which are very rich in the region. This heritage is a potential resource which is not valued enough. Instead, ambivalent phenomena are observable, such as the increase of +160,523.52 m$^2$ in land consumption between 2010 and 2016 (http://goodpa.regione.marche.it/dataset/database-del-consumo-di-suolo-1-10000-1954-1984-2001-2010, accessed on 6 August 2022).

Despite these weaknesses, there are also strength factors. For example, strategies related to art, culture, and creativity have been supported since 2012 with the "Evolved Cultural District" (DCE), a regional project that gave rise to cultural-based innovative initiatives for creative entrepreneurship. Moreover, the SNAI concept of artistic residencies, the "Asili d'Appennino" (lit. Apennine Asylums) promoted the reactivation of heritage buildings and a new concept of hospitality and slow tourism that combines receptivity, culture, and education. Tourism is also a relevant factor of dynamicity for the FA with a +31.46% summer population increase (Government Open Data Regione Marche, 2020), even though an effort should be made to diversify the offers and adjust tourism flows throughout the year.

The analysis, especially the one conducted through qualitative methods, highlighted the need for stronger networks in the FA. Isolated cities, towns, and villages struggle to find common objectives and create new links, even though the impulse of the SNAI led to the implementation of common projects, such as the bike ring path "Ciclovia Alte Marche", linking the nine municipalities, a pilot project that showed the success of such cooperation processes. Yet, associations and local operators have especially raised the issue of establishing and/or strengthening linkages among each other. Even the reopening of the railway between Fabriano and Pergola, promoted by the regional government, struggled to create the consensus that was wished for. In this condition of uncertainty from one side and expectation from the other one, the framework provided by the B4R project has represented a starting point to discuss with local institutions about possible future paths of development and potential new cooperation models to reconnect these centralities through creativity and design.

### 4.5.2. Val Di Sole, Trentino Province: Soaking in the Shapes of Water to Enhance Sustainable, Sensitive, and Slow Tourism

The Val di Sole FA–enclosed in the Brenta Dolomites, the Adamello, and the Ortles-Cevedale Mountain groups–stretches for 40 Km along the shaft of the Noce river and the Vermigliana stream, at an altitude comprised between 680 and 1800 m above sea level, with peaks exceeding 3500 m above sea level. The valley covers a surface of 611 Km$^2$ and it is comprised of 13 municipalities and many small hamlets. A large portion of the territory consists of protected areas: the Adamello Brenta Nature Park in the south and the Stelvio National Park in the north offer important nature reserves for wildlife and plant species in the Alps. Due to the mountainous conditions of the territory and the highly valued and protected natural areas, the 15,600 inhabitants (ISTAT, 2020) are concentrated in the main centers of the valley floor and in the side valleys of Peio and Rabbi. In 2016, the Val di Sole was recognized as the second pilot area for the Autonomous Province of Trento (SNAI 2014–2020). This allows a socio-economic strategic project to be outlined (updated in February 2020) that addresses the sustainable transformation trends for the near future. However, a spatial and landscape design perspective were generally missing as well as an evaluation of the natural capital as operative dynamic resources to cope with natural and anthropic risks. Therefore, the University of Trento focused on the enhancement of nature-based actions in relation to the blue and green environmental systems and to the ecological trails [60].

Etymologically, the name of the valley "Sol" refers to the Celtic water deity *Sulis*, identified by the Romans with Minerva. In fact, the main element that characterizes the Val di Sole is the water, a valuable resource and essential part of the local landscape and community. Water's connection with the territory is found through multiple forms and activities: streams, rivers, waterfalls, snow, glaciers, thermal springs, mills, hydroelectric plants, rivers, and winter sports. Centuries of interaction have produced a fragile but fascinating balance between alpine territory and human needs. Here, the natural and hydrogeologic risks, as well as the anthropic risks, should be addressed through comprehensive, multi-level and trans-valley landscape planning, design, maintenance, and care operations.

Within this framework, the RU of Trento promotes a strategic design approach based on innovation with nature: implementable local actions in the small thermal villages and landscapes can support the preservation and enhancement of the natural capital as the shapes of water. This led to the elaboration of the "Val di Sole Blueprint" territorial portrait (Figure 8) intended as a territorial brand and a tool to promote a new narrative for the valorization of water and thermal landscapes. Beside the technical meaning of "drawing", the term "blueprint" refers to a preliminary strategic plan for future medium- and long-term achievements. The "Val di Sole Blueprint" critically summarizes, shows, and highlights the water resources and their potential for the territory. In doing so, it also supports a sustainable development that connects places for a better quality of life. By making visible the invisible connections [24], the map combines the settlements related to the production (Peio and Rabbi) and the traces of the water resources throughout the territory: this interpretation foresees a critical and conceptual vision as a strategic tool for the spatial and ecological management of the territory.

### 4.5.3. Sicani, Sicilia Region: A Polynuclear and Reticular Settlement System Crossed by Creative Dynamics and New Inhabitants

The Sicani Focus Area (FA) is located in Sicily in a hilly territory halfway between the cities of Palermo and Agrigento, in an area that takes its name from the Monti Sicani mountain. The total settled population is 54,969 inhabitants (2019) and the average population density is 57 inhabitants per square kilometer.

The FA's 18 municipalities are experiencing different trends that have substantially contributed to the peripheralization process: progressive depopulation (–11.11% change in population between 2011–2019), aging population (the average old-age index in 2020

is 266.46 in comparison with the 159 regional value), a progressive abandonment and economic fragility (the average per capita income in the FA is EUR 12,469 compared to the regional value of EUR 15,846), difficult accessibility, and low access to primary services.

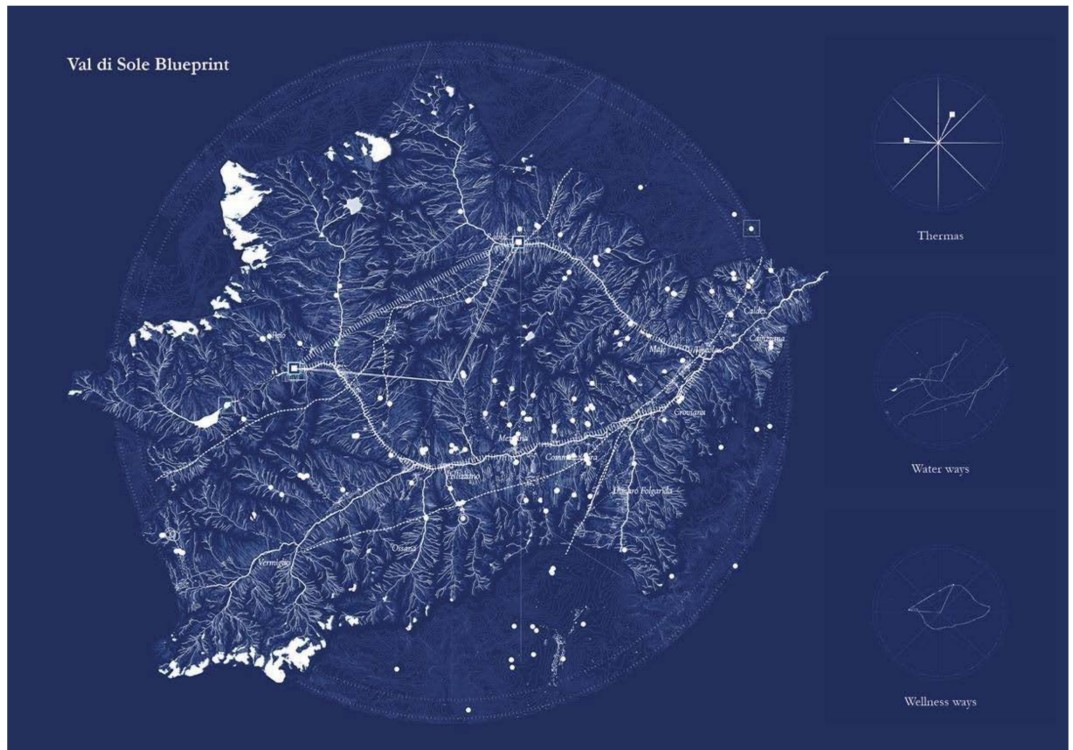

**Figure 8.** The "Val di Sole Blueprint" territorial portrait. Credits: coordination S. Favargiotti, graphic elaboration M. Pasquali, 2021, ©Branding4Resilience, 2020–2023.

Indeed, in the Sicani area both the "SNAI Sicani"—which does not cover the entire FA and includes instead some coastal municipalities—and two local action groups (LAGs) (LAG Sicani and LAG Belice) coexist. Many other projects are ongoing. This dynamism, while revealing an overall liveliness of the territory, highlights a fragmented local governance with variable spaces of interaction between public and private entities and the overlap of a considerable number of director's cabins, programs, and projects at European and national levels.

Despite these data, the analysis highlighted some positive dynamics: experiential tourism models (Sant'Angelo Muxaro and Val di Kam); an important agricultural vocation; a collective brand that identifies the Sicani quality products and tourist attractions (like the "*Distretto Rurale di Qualità dei Sicani*"); a stratified system of natural and cultural resources and abandoned resources which have not yet been sufficiently valorized; experiences of creative activation of the territory (e.g., the case of the Andromeda Theater) and lively local associations; regeneration trajectories (e.g., the "Case a un euro" initiative in Sambuca di Sicilia and the spontaneous repopulation process in Cianciana); a "fascinating" and stimulating atmosphere suitable for those engaged in creative activities and the presence of "neo-ruralists".

The implementation of mapping and the qualitative investigations have returned a spatialized vision of a polynuclear system of small settlements subjected to dynamics of fragmentation and polarization. This is the result of differentiated systems of relations with coastal areas, repopulation trends, and a differentiated endogenous capacity to construct policies to govern territorial resources (Figure 9).

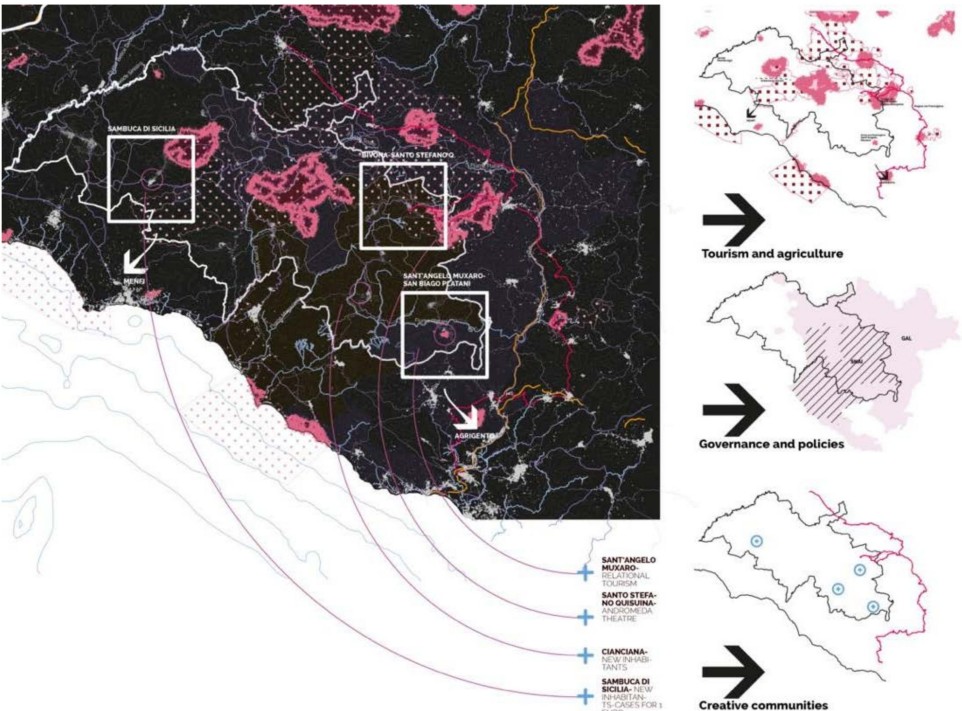

**Figure 9.** The "Sicani polynuclear and reticular settlement system" territorial portrait. Credits: coordination B. Lino, graphic elaboration B. Lino and A. Contato, 2021, ©Branding4Resilience, 2020–2023.

Today, the many centers of the Sicani need to recompose the different transformation trajectories, both the spontaneous ones or the ones carried out by the various existing projects, to find common objectives and to establish new collaborative links [61]: through networking the already existing constellation of people—new settlers, temporary citizens, and travelers—the small centers of the Sicani area can be understood as a premise for a greater cooperation between the coastal area improving the forms of relational tourism, and proposing innovative models of agriculture, entrepreneurial innovation, creative districts, and a different work/life model based on multi-place living, new mobility modes, and digitalization.

### 4.5.4. Alta Valsesia, Piedmont Region: Multiple Tangible and Intangible Interconnections among Mountains, Rivers, and Cultures

Alta Valsesia is an Alpine area in the northern part of Piedmont Region, which takes its name from the Sesia river. The main valley (called Val Grande) extends from Monte Rosa, at the foot of which lies the municipality of Alagna Valsesia, up to Varallo Sesia, classified by SNAI as a pole, which has about 7000 inhabitants and where there are commercial activities and the most relevant schools and services.

The territory of Alta Valsesia includes 17 municipalities (Varallo excluded) and numerous small hamlets. The total settled population amounts to 3515 inhabitants (2019) and the average population density is very low (6.69 inhabitants per square kilometer).

The progressive depopulation and aging population represent one of the main issues of this FA, excluding Alagna Valsesia, whose population and tourism trends both rose in the last few years. Therefore, the built heritage is going to be progressively abandoned or only partially used throughout the year. The settlements of the 17 municipalities consist of villages, with very small centers and numerous hamlets in the proximity, most of which are reachable only by foot. Three different altitude levels characterize the settlement dynamics: valley (villages), middle height (hamlets), and high altitude (mountain pastures).

Despite the fact that many settlements are not easily accessible by car, a dense network of paths connects hamlets and lateral valleys, located in the natural environment. The network of paths and rural chapels, as well as the system of small infrastructures linked to the traditional economy (mountain pastures, productive assets, and bridges, which still have their original materials and construction technologies) are still clearly legible, but in many cases they are characterized by abandonment and decay. However, the natural environment and the traditional built heritage represent possible development potentials for new economic activities linked to slow tourism.

The interconnections across those territories are not only tangible in the natural environment, but also intangible. The historical, architectural, and cultural heritage represent, in fact, an important intangible value; in particular, the ancient Walser tradition and culture join six colonies which were established between the 12th and 13th centuries. Art, vernacular architecture, textile craftsmanship, and wood working represent other key factors that unite ancient cultures, traditions, and people from the different valleys of Alta Valsesia.

The "Exploration" phase highlighted the two lateral valleys (Val Sermenza and Val Mastallone) as very strategic areas to be enhanced, both for their vulnerability and for their values. On the one hand, the accommodations offered are rather limited and tourist flows could be increased during the winter and the summer periods. On the other hand, the two valleys are both characterized by the presence of valuable natural and cultural heritage that, if properly enhanced, are able to attract different target users.

To this aim, it is worth mentioning that in these peripheral areas, characterized by an administrative fragmentation and an increasing depopulation (especially in the youth segment), the territorial network is fundamental for the design and implementation of resilient strategies for reactivation of those places. In Alta Valsesia there are three main networks: the Mountain Union of the Municipalities of Valsesia, the Terre del Sesia Local Action Group (LAG), and the Valsesia Vercelli Tourist Board (ATL).

## 5. Discussion and Conclusions

The exploration of the contexts showed similar patterns but also divergent trends and peculiarities for each context. Each of the FAs are characterized by the presence of high natural capital at risk due to severe natural and anthropic vulnerabilities. However, some FAs (Appennino Basso Pesarese e Anconetano, Val di Sole, and Sicani) show an impactful transformation of the landscape and of the territorial structures that significantly compromise biodiversity and natural resources due to the pressure of anthropogenic changes and urban development. The marginal condition of the Alta Valsesia FA helped to protect the natural resources and today offers a unique condition of balanced coexistence of landscape and urban settlements. The FAs' settlements present a rich variety of tangible and intangible heritage that are not properly enhanced and used for tourist purposes, as well as for innovative residential and productive activities. While the dynamic of the tertiary sector is mainly characterized in all of the FAs by growing touristic trends, the productive sectors show diversified economies based on industrial production (Appennino Basso Pesarese e Anconetano), hydroelectric power (Val di Sole), wood and stone building construction (Alta Valsesia), and co-creative communities (Sicani). The primary sector is actively present in all the FAs by offering a variety of DOC and DOP products (excellence products). In each of the FAs, the territorial governance appears fragmented with a lack of a coherent and multilevel strategic approach able to coordinate actions and projects in different municipalities. This condition often limits the success of design actions or shows the proliferation of numerous active actors that, while promoting social vibrancy and activating cultural and social initiatives, are also sometimes overlapping. The community is fragile, with some innovation trends in place in all of the FAs.

Despite the diversity of many indicators, the application of the proposed methodological approach was useful to explore the analyzed contexts and to outline guiding multidisciplinary themes and development trends for each FA, addressing demographic, social, economic, spatial, settlement, architectural, natural, and infrastructural aspects. The

integration of different qualitative and quantitative tools allowed for the structuring of a more comprehensive portrait of the Italian inner and marginal territories with a wider perspective. The conditions of marginality of the FAs often represented a weakness but at the same time they fostered positive impulses towards enhancement processes. Thus, the achieved results contributed to the advancement of research in this field in different aspects and particularly:

- In terms of definition and knowledge of inner territories in the Italian national context, exploring and understanding the phenomenon of areas suffering territorial imbalances and broadening the Italian research panorama in this field by implementing four empirical case studies that try to go beyond the perspective of SNAI for peripheral territories.
- In terms of methods and tools, experimenting and integrating qualitative and quantitative approaches to the investigation of territories through multidisciplinary perspectives capable of capturing the different conditions of marginality and the different trajectories of transformation at stake in the various contexts, going beyond parameters that lead to standardize and normalize complex phenomena.
- In terms of approach to regeneration and transformation, that should hybridize close looks at territories with large-scale perspectives, in order to implement transcalar visions and strategies that, while shaping and rooting themselves in the specificities of places, establish new collaborations and complementarities between territories.
- In terms of processual and strategic approach, overcoming the opposition between top-down and bottom-up actions, reinterpreting the place-based approach in an even more radical and collaborative way.

With these multiple perspectives, the conditions of marginality that emerged from the "Exploration" phase supported the B4R research units to interact with local and supra-local institutions about possible future paths of development and potential new cooperation models to rethink these areas as new centralities. As briefly discussed below, the results achieved at the local scale became the opportunity to widen the reflection on major and common trends at national level. These can be further investigated both in the next phases of the B4R project and in future research lines on inner territories.

### 5.1. Beyond SNAI: The Marginality of Inner Territories as a Space of Possibilities

The B4R project envisions the possibility of a new role of the peripheral and marginal contexts in relation to metropolitan areas, for a more balanced human-natural lifestyle. To do so, the proposed definition of inner areas does not fit with the proactive complexity of the investigated territories, as it minimizes to a flat surface otherwise rich lands with unique qualities embedded in their natural, human, and socio-economic features. Studies and funding initiatives have mainly focused on economic, services, and infrastructure marginality, basing the classification of the inner areas on quantitative indicators, mainly oriented towards assessing their accessibility to essential services. These places which are considered as marginal or peripheral to main cities, services, and metropolitan economies, account for almost 53% of the municipalities, 23% of the population, and represent about 60% of the entire Italian territory. Based on these data and on the findings collected with the study and comparison of the four focus areas, B4R intends inner territories as "reserves of resilience" [28] that are not residual and require a sensitive, respectful, and sustainable design-driven approach in order to preserve their territorial resources. Here, the value of natural resources, the contemporary interpretation of ancient processes to manage the land, and the innovative practices to take care of landscape and built heritage in vulnerable contexts, are parameters to explore and comprehend the territories as well as indicators for the quality of life.

### 5.2. Transversal and Interdisciplinary Approach of the Research as a Main Result of the Project

The authors and research coordinators propose an interdisciplinary and multiscale approach based on the integration of architectural heritage, natural capital, urban and

territorial planning, and real estate appraisal. The resulting methodological approach is based on the combination of qualitative and quantitative tools, structured according to the multiple competencies and skills that are present in the research team. The ongoing applied research tested an innovative and unique collaborative methodological approach coordinated, shared, and adopted among the four research units. The experiences conducted through this methodological approach and compared among the different FAs and the local stakeholders involved show that, beyond the local territorial specificity, the research unit assumes a coordinating and mediating role to help resolve local conflicts and accompany the design process, in the "Co-design" phase, towards a wider territorial consensus. At the base of the research is a collaborative and participative design and planning action that involves experts and citizens in the fields. By studying and comparing the four inner territories in Italy, we can support local communities with spatial design of tourist infrastructures in selected small villages as agents of a larger transformation path, towards resilient communities and new open habitats. In this framework, we are thus trying to convey a special approach to branding in inner territories, one that goes beyond the definition of a territorial marketing strategy, placing the project at the center. Indeed, branding in B4R connects to the broad and multidimensional role of the project as a collaborative expression of purposes and as a means for envisioning integrated future scenarios.

### 5.3. Themes and Topics Raised from the Transversal Reading of the Four Territories

Increasing accessibility in terms of mobility and a greater supply of services, particularly those for health and education, as well as the digital infrastructure, are necessary and urgent actions to revive the development trajectories of these territories. However, to achieve sustainable transformations through these strategies, it is necessary to set in motion an economic and social model that is capable of reactivating latent resources, retaining or attracting population and proposing competitive models of life compared to those offered by large urban areas.

Emerging designs and project themes developed during the "Exploration" have highlighted potentials and resources specifically related to space, infrastructures, settlements, and landscapes that can address the challenges of the current marginal condition. With an approach related to the capacity of design of grasping intuitively and perceptively the quality of a place, these design explorations have been helpful to focus on characteristics, specificities, and criticalities of each territory.

The B4R project will further investigate how strategies, policies, and projects can contribute to define new alliances with bigger cities and coastal areas, to offer more attractive habitat conditions to local communities and to potential new inhabitants, while enabling innovation and creative processes.

The strengthening of digital networks on the national territory is necessary on several fronts: it can fill some gaps in accessibility to services and reduce the demand for mobility, it can facilitate distant learning and a more diffused use of telemedicine, and it can support remote work. In particular, the acceleration in the diffusion of ICT imposed by the pandemic leads one to think that, in the near future, remote work will be a viable solution for some workers and a real opportunity to improve the attractiveness of inner areas by offering them as new places to live. With remote work, not only working times but also working spaces are changing, and this requires new settlement and spatial models and dedicated services.

The opportunities related to built heritage, natural resources, and human capital can be more effectively linked to forms of relational tourism, through networking, slow mobility models, and the implementation of minimal tourist infrastructures, but also to agriculture and entrepreneurial innovation, to creative districts, thanks to urban policies that encourage spillovers and spin-offs. The special constellation of people already present—new settlers, temporary citizens, and travelers—suggests that the tourism of the future will be able to merge with different models of work/life based on multi-place living, new mobility, and digitalization.

All of these considerations have made clear the need to address challenges in a multi-faceted way. The operative branding actions that B4R started to envisage with the different stakeholders of the four FAs face urgent challenges for each place, but at the same time aim to become paradigmatic cases, useful for other contexts. In all of the FAs, the branding strategy—in the sense that we tried to convey in this article, namely as a strategy directly connected to the transformative potential of territories and to the capacity of spatial design to become a significant asset and agent for the territory—has shown the potential to empower communities with coordinated interventions that represent a viable pathway towards a major resilience, through creation of jobs, increased visibility and attractivity, positive economic rebound effects, and the creation of new metabolisms. An ex-post evaluation of the impacts of these branding actions in the different FAs is a goal and a future perspective for the continuation of the B4R project.

*5.4. Limitations*

The proposed methodological approach for the exploration of the selected inner territories was developed to be applied in different territorial contexts and its flexibility allows researchers to apply it simultaneously in different areas. Nevertheless, it is worth mentioning that the results achieved from the application of its steps are not always easily comparable for several reasons. First of all, the selection criteria led to the identification of four fragile inner territories, characterized by common phenomena (such as aging population, lack of essential services, decreasing population, and tourism trends), but with very heterogeneous territorial and environmental contexts, which obviously reflect different enhancement strategies and governance models. This implies that, as a matter of fact, the "Exploration" phase needs to be partially adapted on the basis of the territorial specificities. In particular, a great effort has been spent to coordinate the maps' contents with specific regard to the challenge of data availability and data updating. Often, the RUs experienced dissimilar advancements in the GIS planning systems of the four regions, a fact that created also slow downs depending on the usable dataset at the regional and local levels. Some Italian regions have numerous open data, organized in GIS and freely accessible by means of dynamic geoportals, while in other regions the available datasets are limited and not recently updated. Moreover, the number and typology of stakeholders can vary significantly and, as a consequence, differently influence both the exploration and the access to information as well as potentially influence the following co-design and co-visioning phases. The possibility to create comparable datasets and maps derives from the strong will to coordinate the work despite these challenging differences and to deliver a comparable and scientific pool of information that is useful to perform comparative analyses as a basis for next project phases.

The exploration of the selected inner territories should also be supported by research on the field, visiting and knowing the places, dialoguing with people and local actors, and direct living experiences in the villages and in the natural environment. This approach was extremely difficult at the beginning of the B4R project due to the COVID-19 pandemic, which limited the on-site research and delayed the whole exploration process.

Finally, one limitation of the approach stands in its richness: different disciplines and qualitative and quantitative approaches are interrelated and thus necessitate coherent and constant coordination. In this regard, a limitation stands in the difficulty of integration and transversality of the multiple disciplinary fields involved in the project, which risks being vanished if not properly managed.

*5.5. Future Developments: Co-Design and Co-Visioning*

The analyses conducted in the "Exploration" phase of the project have made it possible to highlight the stratification of tangible territorial resources (natural areas, cultural assets, infrastructures, historical centers, general services, production of excellence, etc.) and intangible resources (knowledge, local traditions, and forms of innovation), and to identify some of the fundamental factors that have contributed to the process of peripheralization

of the areas (physical and digital disconnection, economic and social fragility, depopulation and aging). The final aim of the project is to envisage spatial concepts that denote specific qualities to provide interpretative frameworks of spatial structure and of future spatial developments. To achieve this, the results of the analytical phase collected in this article are essential.

To lead inner territories of the four FAs to a condition of higher resilience, B4R puts in place participatory processes through four "Co-design" workshops to test specific solutions for each context with the communities. The four workshops (Sicily, Marche, Trentino, Piedmont) have been conceived as a way to approach territories through project explorations, disclosing central issues posed by the research theme, which are verified and deepened through design. Indeed, during the co-design workshops, transformations and minimal interventions on the urban and territorial context are summarized in project proposals. Each workshop produces proposals delving into the specificity of each area. The projects address key issues regarding the contexts' future development, in agreement with the local administrations, and developed by a pool of expert designers and researchers who in one day propose a vision of change in response to the requests coming from the area. The whole process contributes to increase community's awareness on places, to promote larger strategic networks, and to take care of the analyzed territories. The co-design workshops are an opportunity to imagine the reactivation of potential spaces as catalysts of new energies and flows. This ultimately leads to the creation of a shared identity, and at the same time provides useful work ideas for other municipalities in the FA. In fact, thanks to the debate with local actors, to the listening of local stories, to the physical exploration of places, and to the exchange between participants, each workshop managed to look into the paradigmatic nature of each pilot case (PC) identifying possible branding operative actions that, even when focusing on the specific contextual conditions, are scalable and adaptable responses to the entire FA, as well as potentially significant answers for the other FAs involved in the project. The results of these collaborative workshops are currently being discussed and critically formulated by the four RUs in dialogue with the local communities. Therefore, they are not part of the results presented in this article. They will serve instead to inform the last step of the project.

In the "Co-visioning" phase B4R will look at a possible horizon of transformation and development to address the FA's challenges and issues. The project will propose a process of strategic spatial co-visioning to recompose in a coherent territorial vision the current system of relationships, identifying a branding strategy that can unveil and enable the territorial capital in all its dimensions (natural, cultural, built, financial, social, institutional) to increase the communities' transformative capacity. The branding strategy will try to achieve a coherent spatial vision of development to increase the interaction capacity of the area, enhancing and connecting material and immaterial heritage and resources. In other words, a new development path able to enhance the value of the different territorial capitals through design.

**Author Contributions:** This paper is to be attributed in equal parts to the four authors (M.F., S.F., B.L., D.R.), who have collaboratively contributed to its conceptualization, methodology, and data curation; operatively, the writing—original draft preparation—, review, and editing for each paragraph have been managed as it follows: Sections 1 and 2.1 M.F., S.F., B.L. and D.R.; Section 2.2 M.F.; Sections 3 and 4.1 M.F., S.F., B.L. and D.R.; Section 4.2.1 M.F.; Section 4.2.2 S.F.; Section 4.2.3 M.F.; Section 4.2.4 D.R.; Section 4.2.5 B.L.; Section 4.3 M.F., S.F., B.L. and D.R.; Section 4.4 M.F.; Section 4.5 M.F.; Section 4.5.1 M.F.; Section 4.5.2 S.F.; Section 4.5.3 B.L.; Section 4.5.4 D.R.; Section 5 M.F., S.F., B.L. and D.R. Supervision, M.F.; overall project administration, M.F.; local project administration, M.F., S.F., B.L. and D.R.; funding acquisition, M.F., S.F., B.L. and D.R. All authors have read and agreed to the published version of the manuscript.

**Funding:** "B4R–Branding4Resilience. Tourist Infrastructure as a Tool to Enhance Small Villages by Drawing Resilient Communities and New Open Habitats" (Project number: 201735N7HP) is a research project of relevant national interest (PRIN 2017—Youth Line) funded by the Ministry of University and Research (MUR) (Italy) for the three-year period 2020–2023. The project is coordinated by the Università Politecnica delle Marche, with Principal Investigator Prof. Arch. Maddalena Ferretti, and involves as partners the Università degli Studi di Palermo (Local Coordinator Prof. Arch. Barbara Lino), the Università degli Studi di Trento (Local Coordinator Prof. Arch. Sara Favargiotti), and the Politecnico di Torino (Local Coordinator Prof. Arch. Diana Rolando).

**Institutional Review Board Statement:** Ethics approval is not required for the present study.

**Informed Consent Statement:** Informed consent was obtained from all subjects involved in the study.

**Data Availability Statement:** The data that support the findings of this study are derived from on field research and resources available in the public domain.

**Acknowledgments:** The authors would like to thank all the B4R researchers for their work, contribution, and support in the project. In particular we would like to thank the researchers that have developed the exploration activities, the platform, the communication, and visual identity.

**Conflicts of Interest:** The authors declare no conflict of interest.

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
