# Peer review of "Branding4Resilience: Explorative and Collaborative Approaches for Inner Territories"

_sustainability, doi:10.3390/su141811235_

Round 1

Reviewer 1 Report

Review

The aim of the article is to look at the future of the inner territories in four Italian peripheral contexts by proposing a methodological approach and outlining and discussing the first results and future stages of a project entitled 'B4R Branding4Resilience', funded by the Italian Ministry of Research from 2020 to 2023.

The assumed goal is interesting for defining the role and position of local self-government in the context of national policies. The aim of the study was accomplished on the basis of the conducted research. The article focuses on qualitative and quantitative approaches and the first few results. The results synthesize the main findings of exploration in four selected thematic areas. They outline the reading of territories through data analysis and mapping, perceptual narrative explorations, field studies, and exploration projects. It is the right approach to the undertaken issues, which have been precisely described in this work and I rate it very highly substantively.

The discussed issue of peripherality is an important topic of regional research in many countries that are historically advanced in the implementation of local management and have achieved success by implementing an interdisciplinary strategy of strengthening the role of regions in creating innovative economy through the use of innovative management methods. The proposed branding strategy suggests a double view, internal through the participation of the local community, and an external one, aimed at increasing the attractiveness of the areas as a result of providing services for relational and experiential tourism. The discussion on this topic has been intensified due to problems with the effects of Covid. The pandemic in all its severity highlighted territorial differences, especially in the field of health and educational services, from which internal areas are by definition distant, and revealed the limits of the constant urbanization and concentration of settlements and infrastructure policy in large urban agglomerations.

The issues raised are extremely important for studying the role of regions in the development process of peripheral areas and may be a significant proposition for regions with a lower level of integration than the studied areas. The practical dimension of the research carried out and the results obtained is a good example of the transfer of knowledge to practice in the field of spatial planning, innovative management and the benefits of cooperation between science and practice. I rate the article very highly, I propose to publish it in an unchanged form and content.

Author Response

We thank the reviewer for the generous appreciation and the comments.

Reviewer 2 Report

This topic is quite interesting, but the paper is far away from a research article, instead, it reads more of a report. The authors need to focus on the first part of your research proposal rather than the entire research plan.

Another serious issue is that the write style manifests a research beginner's work that lacks of training in academic writing. For instance, strong arguments should be backed up by citations. Introduction should introduce research background and throw hook to appeal audience. In contrast, this introduction is pretty dry and unclear about writing purpose. 

Last but not the least, there lacks of a rigorous methodology. The research framework is too vague and is hard to get what is your main point. The methodology section lacks of elaboration on the first part of your research to lay out the research tools and steps employed in the study. 

Please use complete sentences. 

Reviewer 3 Report

The study is very innovative and the research questions have been addressed in full. The methodological approach is robust and the findings very interesting for the academic and managerial implications.

Author Response

(The authors gave the same response as above.)

Reviewer 4 Report

This paper could present some ideas about B4R Branding4Resilience. However, the links to the extant literature and the study contribution are not strong enough in their current state. One of the main criticisms is that the actual contribution of the paper is unclear. There needs to be a more apparent articulated contribution to these concepts.

The authors have done well citing recent literature. In a potential revision, authors should use more citations from 2020 to 2022 papers to emphasize the relevance of their study and support and strengthen their arguments.

The authors are silent about the adoption of the particular methodology. Would you please provide proper arguments with references?

In the discussion section, please compare your findings with earlier studies (converge or diverge)? Although the authors highlight the research contributions, these should be captured under the implications section to promote understanding.

The paper has significant challenges with the brevity and clarity of communication, which is unclear. Grammatical and spelling checks should be done. This is very critical and significant in enhancing clearer communication.

Round 2

Reviewer 2 Report

The manuscript is excessively long. It would be better to shorten the paper to make it flow better and clearer. 

Author Response

We thank you the reviewer for the additional comments. 

We have shortened the article, working particularly on the results, in paragraphs 4.4, 4.5.1.–4. We have also revised entirely the English language and style to address the requested minor spell checks.

We are confident that with these additional corrections the article is more readable and overall clearer.